

2

**Lightning Assimilation in the Weather Research and Forecasting (WRF) Model Version 4.1.1: Technique Updates and Assessment of the Applications from Regional to Hemispheric Scales**

Daiwen Kang[1*], Nicholas K. Heath[2#], Robert C. Gilliam[1], Tanya L. Spero[1], and Jonathan E. Pleim[1]

[1] Center for Environmental Measurement and Modeling, Office of Research and Development,

9          U.S. Environmental Protection Agency, Research Triangle Park, NC 27711, U.S.A.

[2] Air Quality and Atmospheric Composition, Atmospheric and Environmental Research,

11         Lexington, MA  02421, U.S.A.

*Correspondence: kang.daiwen@epa.gov
#Currently working as an independent consultant





**Abstract:** The lightning assimilation (LTA) technique in the Kain-Fritsch convective

parameterization in the WRF model has been updated and applied to continental and hemispheric

simulations using lightning flash data obtained from the National Lightning Detection Network

(NLDN) and the World Wide Lightning Location Network (WWLLN), respectively. The impact

of different values for cumulus parameters associated with the Kain-Fritsch scheme on

simulations with and without LTA were evaluated for both the continental and the hemispheric

simulations. Comparisons to gauge-based rainfall products and near-surface meteorological

observations indicated that the LTA improved the model's performance for most variables. The

simulated precipitation with LTA using WWLLN lightning flashes in the hemispheric

applications was significantly improved over the simulations without LTA when compared to

precipitation from satellite observations in the Equatorial regions. The simulations without LTA

showed significant sensitivity to the cumulus parameters (i.e., user-toggled switches) for monthly

precipitation that was as large as 40% during convective seasons for month-mean daily

precipitations. With LTA, the differences in simulated precipitation due to the different cumulus

parameters were minimized. The horizontal grid spacing of the modeling domain strongly

influenced the LTA technique and the predicted total precipitation, especially in the coarser

scales used for the hemispheric simulation. The user-definable cumulus parameters and domain

resolution manifested the complexity of convective process modeling both with and without

LTA. These results revealed sensitivities to domain resolution, geographic heterogeneity, and the

source and quality of the lightning dataset.

## 1. Introduction

Thunderstorms are natural phenomena that have intrigued human imagination for

thousands of years. Although early efforts in atmospheric science and modeling were focused on



understanding and forecasting thunderstorms, they remain difficult to accurately simulate in
meteorological models. A variety of lightning parameterization schemes have developed in
regional and global atmospheric models (Price and Rind, 1992; Romps et al., 2014; Finney,
2014; Lopez, 2016) based on various physical, dynamical, and cloud properties, but these
schemes marginally reproduce the spatial and temporal variability of lightning flashes with
varying success over different regions of the globe. With the advancement of lightning detection
technologies both at ground level and via satellite in the past decades, observed lightning flashes
with coverage from regional to global scales are available and can be used for lightning
assimilation (LTA). A robust LTA can improve convective simulations in meteorological models
for retrospective atmospheric simulations (e.g., Heath et al., 2016; Marchand and Fuelberg,
2015) or help generate better initial fields for real-time weather forecasting (e.g., Lagouvardos et
al., 2013; Giannaros et al., 2016; Fierro et al., 2012, 2015) by pinpointing where deep convection
occurred and altering the meteorology in what is generally referred to as a hot start (Gan et al.,

2021).

Heath et al. (2016) implemented an LTA technique in the Kain-Fritsch (KF) convective

scheme in the Weather Research and Forecasting (WRF) model using lightning observations
from the National Lightning Detection Network (NLDN) over the contiguous United States
(CONUS).  They found that the simulation of warm-season rainfall was substantially improved,
and other near-surface meteorological variables were clearly improved in retrospective WRF
applications. Lightning also profoundly impacts the chemical composition of the troposphere by
generating and releasing nitrogen oxides ($LNO_x$) that can significantly alter ground-level ozone
($O_3$) concentrations in some regions (Kang et al., 2020). Because meteorological models drive air
quality simulations, improving meteorological variables with LTA will cascade to chemistry



fields simulated by air quality models. It is especially critical when $LNO_x$ emissions are included
in air quality models, since LTA is designed to align $LNO_x$ emissions with the time and location
when atmospheric convection occurred in the model, so the subsequent chemistry reactions and
transport will more accurately reflect the emissions from lightning (Kang et al., 2019a and
2019b).

Heath et al. (2016) implemented the LTA technique in WRFv3.8 and tested for several

month simulations. The LTA technique has been implemented in subsequent WRF releases (not
publicly available yet) and applied in many meteorology and air quality studies over the CONUS
(e.g. U.S. EPA, 2019; Appel et al, 2021). Although using LTA improved the predicted
meteorological variables, some occasional unwanted departures from base model predictions
without LTA occurred. Most commonly, LTA resulted in a low bias in summertime rainfall in
some regions (U.S. EPA, 2019).

For this reason, it is of interest to investigate two parameters associated with the KF

convective scheme with different optional values, which are specified in the WRF runtime
namelist input file, are often encountered by WRF users
([https://www2.mmm.ucar.edu/wrf/users/docs/user_guide_v4/contents.html](https://www2.mmm.ucar.edu/wrf/users/docs/user_guide_v4/contents.html)) . One parameter is
called kfeta_trigger (also referred to as trigger for simplicity in this paper) which controls the
conditions to determine how the KF convective scheme is triggered with three optional values: 1,
the default value; 2, moisture-advection based trigger (only for ARW - the advanced research
WRF dynamical solver); and 3, RH-dependent additional perturbation to Option 1 (not tested).
Another parameter is called cudt (namely **cu**mulus time interval, **d**elta **t**) and its value determines
the minutes between cumulus physics calls (here it is the KF scheme). The default value of 0
indicates that the cumulus physics is called at every model step, and any non-zero value specifies



the interval (minutes) that the cumulus physics is called (for example, cudt=10 means that the
cumulus physics is called every 10 minutes). Even though some discussions and
recommendations regarding the choice of these parameter values through online forums or WRF
user mailing list (e.g., https://forum.mmm.ucar.edu/; https://wrfems.info/;
https://www.epa.gov/sites/default/files/2017-02/documents/wrf_with_ltga_userguide.pdf), but no
literature evaluates how these parameter values impact model performance when LTA is used.

Heath et al. (2016) demonstrated that the LTA technique consistently improved the

simulation of precipitation and other near surface variables, but the evaluation was limited to the
CONUS, reflecting the areal coverage of NLDN (Murphy et al., 2021). As the spatial
applications of atmospheric composition modeling are expanded from regional to hemispheric
and global scales and new lightning datasets are available, there is a strong need to examine how
this LTA technique performs at these larger scales when lightning flash data from a less accurate
detection network are used. Thus, lightning flashes from the World Wide Lightning Location
Network (WWLLN, operated by the University of Washington: http://www.wwlln.com) is a
suitable candidate because it has the global coverage with affordable cost, albeit its detection
efficiency is lower than the >95% of NLDN (Abarca et al., 2010).

Our research has multiple objectives based on the aforementioned open research needs:

1) assess the impact of the parameter values associated with the KF convective scheme on WRF
performance over the CONUS domain without LTA (BASE case) and with LTA using lightning
flashes from NLDN; 2) examine the LTA in WRF using lightning flashes from WWLLN and
compare to the simulations with NLDN lightning flashes; and 3) apply LTA to WRF simulations
over the Northern Hemisphere and evaluate the performance in terms of precipitation and near-
surface meteorological variables. In section 2, we describe the updates made to the initial LTA





technique (Heath et al., 2016). Section 3 provides the detailed data and methodologies of the
model simulations and their evaluation. Section 4 presents our analysis on the impact of
parameters with KF convective schemes with and without lightning assimilation over CONUS
using lightning flashes from NLDN and WWLLN. In section 5, we analyze the use of lightning
flashes from WWLLN for LTA and evaluate WRF simulations with and without LTA over the
Northern Hemisphere. And we conclude with key findings and recommendations in section 6.
**2.  Updates on the LTA technique**

The lightning assimilation used here is based on Heath et al. (2016), which extended the

works of Rogers et al. (2000), Mansell et al. (2007), Lagouvardos et al. (2013), and Giannaros et
al. (2016).  In general, the lightning assimilation approach used here is straightforward,
activating deep convection where lightning is observed and only allowing shallow convection
where it is not.  This method is applied in the Kain-Fritsch scheme in WRF (Kain, 2004).  A full
description of the method can be found in Heath et al. (2016).  Here, we provide only the
essential details, along with recent modifications to the scheme.

First, the lightning data (WWLLN or NLDN) is binned to the WRF domain in both time

and space.  The temporal binning is done every 30 min and includes lightning data from -10 min
to +20 min of the current time.  The spatial regridding searches for a lightning strike within each
grid box (using the staggered grid edge coordinates) within each time bin.  This process creates a
new lightning file with the same horizontal dimensions as the WRF domain filled with zeros (no
lightning) or ones (lightning) at each 30-minute time step.   During the WRF simulation, if
lightning is present, the scheme first goes through its standard updraft calculations, except that it
uses the layer with the greatest moist static energy as its updraft source layer (USL).  If the
resulting cloud does not meet the criteria for deep convection, 0.1 g kg-1 of water vapor and 0.1



K are incrementally added to the USL until deep convection is forced. In the original Heath et
al. scheme, only moisture was added to the USL. We have included temperature perturbations to
further promote activating deep convection in these grid points with lightning.
In the unmodified KF scheme, a cloud must exceed a minimum depth (as a function of
cloud base temperature) to satisfy the deep convection criteria. Heath et al. (2016) modified this
depth for lightning assimilation to be more consistent with lightning-producing storms.
Specifically, within WRF, storms with a base temperature greater than or equal to 20ºC must
have a cloud depth of at least 6 km with a cloud top temperature less than -20ºC. Similarly, in
the original model in Heath et al., storms with a cloud base temperature less than 20ºC must have
a cloud depth of at least 4 km and a cloud top temperature less than -20ºC. These criteria were
set to ensure that sub-grid deep convective clouds were deep enough to have a mixed-phase layer
to support lightning (e.g., Mansell et al., 2007; Bruning et al., 2014; Preston and Fuelberg, 2015).
In this study, we slightly modified the scheme to require that the cloud top is at least one model
level above the -20ºC level, ensuring cloud-top temperatures are less than -20ºC (e.g.,
Stolzenburg and Marshall, 2009). The prior limit at -20ºC could inadvertently weaken simulated
deep convective clouds, which may contribute to the dry bias in earlier applications of lightning
assimilation approaches (U.S. EPA, 2019).
In Heath et al. (2016), if deep convection could not be achieved after incrementally
adding up to 1 g kg-1 to the USL (which is now 1 g kg-1 and 1 K in our update), then no further
action was taken, and deep convection was not activated by KF. However, to increase the
realism of the scheme and increase the odds of deep convection the next time the scheme is
called, we have updated the approach as follows. If a deep convective cloud cannot be activated,
the tallest cloud created is passed into the KF shallow convection scheme. In the KF scheme,



shallow clouds are re-diagnosed each time the scheme is called. For example, suppose a shallow
cloud is generated at t=0 and KF is called at 5 min intervals. In that case, at the t=5 min call, KF
would determine if a shallow cloud is still present. Thus, the cloud can evolve so that at t=5 min
it could have slightly different characteristics than the one diagnosed at t=0. This allows shallow
clouds to grow, decay, or persist at short timescales.
Therefore, if the LTA method cannot trigger deep convection, the shallow cloud that is
generated within WRF can precondition the atmosphere, thus increasing the likelihood of deep
convection the next time the KF scheme with LTA is called. Therefore, these refinements to the
LTA scheme in KF more closely replicate how convective initiation is observed in nature, where
shallow cumulus and congestus clouds precondition the environment prior to deep convection
initiation.
Lastly, at grid points without observed lightning, deep convection is suppressed in WRF,
and only the shallow portion of KF is allowed to run. Because convective clouds in nature can
form and precipitate without generating lightning, this suppression technique serves as a realistic
approach to reproduce nature given the constraints of the KF parameterization.

**3. Data and Methodology**
3.1. Lightning flash data
Lightning flash data from two ground-based lightning detection networks were used for the
assimilation using the LTA technique in this study. The NLDN provides cloud-to-ground
lightning observations with a detection efficiency of >95% and a location accuracy of about 150
m (Murphy et al., 2021) over the contiguous U.S. (CONUS). The WWLLN provides global





lightning data with lower detection efficiency and location accuracy (Abarca et al., 2010;
Rudlosky and Shea, 2013; Burgesser, 2017) compared to NLDN and the Lightning Imaging
Sensor (LIS) observations (Mach et al., 2007). Since WWLLN has global coverage, even with its
relatively lower detection efficiency and location accuracy compared to NLDN, it could be a
good option for applications beyond CONUS. Figure 1 shows how the average lightning flash
rate (flashes km$^{-2}$hr$^{-1}$) from WWLLN compares to NLDN during July and September 2016 when
hourly lightning flash counts are gridded into the CONUS 12-km grid cells.
As shown in Figure 1, the lightning flash rates in NLDN are much higher than those in
WWLLN, especially during July and over the land, and this is generally true (not shown) that
NLDN reported more lightning flashes than WWLLN during warm months over land. The
differences are much smaller during cool months and over the coastal regions where NLDN has
coverage. Note that the absolute difference in flash count may not necessarily translate
proportionally into convective activities in terms of LTA because the LTA technique as
described in Heath et al. (2016) depends on the detection of lightning occurrence (binary "yes"
or "no" situation), not the actual flash count, in a specific time interval at a grid cell.
3.2. Precipitation Data
The daily precipitation from the Parameter-elevation Regressions on Independent Slopes
Model (PRISM)'s high-resolution spatial climate data for the United States
(https://climatedataguide.ucar.edu/climate-data/prism-high-resolution-spatial-climate-data-
united-states-maxmin-temp-dewpoint) is used to evaluate WRF-simulated precipitation over the
CONUS, and the NOAA Climate Prediction Center (CPC)'s global unified gauge-based analysis
of daily precipitation (https://psl.noaa.gov/data/gridded/data.cpc.globalprecip.html) product is
employed to assess WRF's hemispheric precipitation predictions. The daily total PRISM




precipitation data are available at 4-km horizontal grid spacing over the CONUS, and the annual
CPC precipitation (partitioned into daily totals) is available globally at 0.5° latitude × 0.5°
longitude grid (720 × 360) resolution. These datasets were regridded to the WRF modeling
domains for the 12-km CONUS and the 108-km Northern Hemisphere to pair with model
simulations in time and space. To assess the simulated precipitation over the oceans, especially
in the tropical regions where no gauge-based measurement is available, products from the Global
Precipitation Measurement (GPM) (Huffman et al., 2015; Asong et al., 2017), a joint mission co-
led by NASA and the Japan Aerospace Exploration Agency (JAXA) and comprised of an
international network of satellites that provide the next-generation global observations of rain
and snow, are employed. The Integrated Multi-satellitE Retrievals for GPM (IMERG) Long-term
Precipitation Data Products
(https://arthurhouhttps.pps.eosdis.nasa.gov/gpmdata/YYYY/MM/DD/imerg/; registration is
required for access) cover the entire globe with 0.1° latitude × 0.1° longitude grid resolution. To
compare with WRF simulated hemispheric precipitation, the daily mean precipitation data from
the IMERG V06 dataset from 2016 is regridded onto the hemispheric WRF domain
(https://gpm.nasa.gov/data/directory). The research-quality gridded IMERG V06 dataset Final
Run product estimates precipitation using quasi-Lagrangian time interpolation, gauge data, and
climatological adjustment.
3.3. Ground-Based Meteorological Data

The impacts of user-definable parameter values associated with KF and datasets for LTA

were quantified for simulated near-surface meteorological variables such as precipitation, 2-m
temperature (T2), water vapor mixing ratio, wind speed and wind direction. The simulated
meteorological fields from WRF are compared against observations from NOAA National



Centers for Environmental Information (NCEI) land-based stations, which are archived from
data collected globally (https://www.ncei.noaa.gov/products/land-based-station). The
Atmospheric Model Evaluation Tool (AMET) (Appel et al., 2011) is used to pair surface
observations with model predicted values in both space (bilinear interpolation) and time (hourly).
3.4. Model Configurations and Simulation Details

The WRF model (Skamarock and Klemp, 2008) version 4.1.1 (WRFv411,

https://github.com/wrf-model/WRF/releases/tag/v4.1) with LTA updates to Heath et al. (2016)
(as described in Section 2) is used to perform simulations over the CONUS and the hemispheric
domains. The CONUS domain is configured with 36 vertical levels and 12-km horizontal grid
spacing with 472 × 312 grid points. The hemispheric domain is configured with 45 vertical
levels and 108-km horizontal grid spacing with 200 × 200 grid points that covers the entire
Northern Hemisphere and the northern border of the Southern Hemisphere along the Equator.
The simulation period for CONUS simulations is from April–July in 2016 with 10-day spin-up
period from March 22; for the hemispheric domain, annual simulations for 2016 are performed.
Our analysis focuses on July when convective activities are often the most prevalent over the
CONUS; other months are examined in the hemispheric simulations which simulate the year-
round convective activities in the tropics. The detailed configurations of cloud microphysics,
land surface parameters, radiation schemes, and four-dimensional data assimilation (FDDA) are
the same as described in Heath et al. (2016) and sample WRF namelist input files for both the
CONUS and hemispheric simulations are included in the supplementary information (Table S1
and Table S2).

The KF scheme includes two options to trigger convective activity. Trigger 1 is based on

a mass-conservative cloud model, which includes parameterized moist downdrafts, entrainment,



and detrainment at the cloud edge (Kain and Fritsch, 1990, 1993) and allows interaction between
cloud and environment, and it is the default option for most applications. Trigger 2 is an alternate
option based on Ma and Tan (2009), and that is a moisture-advection modulated trigger function
to improve results in subtropical regions when large-scale forcing is weak. In addition, the KF
scheme is called by default at every time step, but it can be configured to only update convective
parameters on a user-definable time increment. In this study, sensitivities are conducted to the
version of the KF trigger (i.e., Trig1 and Trig2, abbreviated as K1 and K2 in Table 1,
respectively), as well as to frequency at which KF is called (i.e., "cudt"). Two sensitivities on
cudt were performed: one where KF is called at each model integration time step (i.e., "Cudt0",
abbreviated as C0 in Table 1), and the other where KF is updated every 10 minutes of integration
time (i.e., "Cudt10", abbreviated as C10 in Table 1). The sensitivities to KF trigger and update
frequency are combined in a matrix of simulations that also are conducted with/without LTA,
and they are listed in Table 1. All eight simulations are performed for both the CONUS and the
hemispheric domains. For LTA cases, lightning flashes from both NLDN and WWLLN are used
over the CONUS domain and lightning flashes from WWLLN are used for the hemispheric
domain. For convenience of description, the cases without LTA are collectively referred to as
BASE cases, and the cases with LTA are referred to as LTA cases. To further distinguish the
lightning networks, the LTA cases are also referred to as LTA NLDN (or simply NLDN) and
LTA WWLLN (or simply WWLLN) cases, respectively.
3.5. Evaluation Methodologies

The assessment of the impact of LTA on model performance is focused on precipitation

since that is the most affected variable, though other near-surface variables are also evaluated.
Due to the highly heterogeneous nature of thunderstorms and lightning over space, in addition to





examining the overall statistics across the modeling domain, statistics are analyzed to assess the

impact of LTA over U.S. climate regions (https://www.ncei.noaa.gov/monitoring-

references/maps/us-climate-regions) in both domains and some of the larger countries in the

hemispheric simulations. Figure 2 shows these climate regions over the CONUS modeling

domain and the selected countries (also referred to as regions) in the hemispheric modeling

domain.

The statistical metrics in this analysis include the widely used correlation coefficient (r)

to measure the linear association of measured and simulated variables, mean bias (MB) and

normalized mean bias (NMB) to quantify the departure of simulated values from measured

values, and root mean square error (RMSE) and normalized mean error (NME) to elucidate the

errors associated with model simulations. More emphasis is placed on certain metrics than others

depending on the nature of the simulated quantity. For instance, with precipitation, correlation

coefficient (if the model can simulate rainfall at the right time and location) and MB and NMB

(if the model over- or under-estimate rainfall amount) are more straightforward than the error

metrics (though they are still relevant), but MB and NMB are inappropriate to evaluate wind

directions.

## 4. CONUS WRF Simulations

As shown in Table 1, four BASE (without LTA) cases, four LTA cases using lightning flash

data from NLDN, and four LTA cases using lightning flash data from WWLLN over the

CONUS domain were performed using the combinations of two trigger options and two

convective update (cudt) intervals, respectively. For the LTA cases, when lightning flashes were

not present, the ShallowOnly option (Heath et al., 2016) was used (Table S1).





4.1. Precipitation

Figure 3 displays the July 2016 mean statistics generated by pairing the gridded WRF

precipitation with the values from PRISM in time and space for each of the U.S. climatological

regions. As shown in Figure 3, the BASE simulations present the most dramatic fluctuations

among cumulus parameter sensitivities than the LTA cases. With Trig1, when the cudt is

changed from 0 to 10, the correlation coefficient is substantially reduced across all the regions

(Figure 3a), and increases in biases (overestimate of precipitation, Figures 3b&c) and errors

(Figures 3d&e) are also worsened by less frequent cumulus updates. With trigger 2, the biases

(MB and NMB) changed from overestimation to underestimation, and the errors (RMSE and

NME) were smaller compared to Trig1. Though the setting for cudt altered simulations with

Trig2, the difference was smaller than the cases with Trig1. In general, the Trig1 cases tended to

produce more precipitation (overestimate compared to PRISM precipitation) than the Trig2 cases

(underestimate compared to PRISM precipitation), and the Cudt10 cases generated more

precipitation than the Cudt0 cases. Among the four cases in the BASE model simulations, the

K1C0 case (Trig1, Cudt0) is the most favorable in terms of the correlation coefficients and

precipitation biases, but the error statistics, especially NME, may not be the most desirable.

Using LTA (Figure 3), the correlation coefficients significantly increased over the

domain and across the regions (from the range of ~0.25 to ~0.40 to the range of ~0.30 to ~0.48)

relative to the BASE cases. Though the LTA WWLLN cases had lower correlation compared to

the LTA NLDN cases due to the lower detection efficiency of lightning flashes in WWLLN, the

improvement was still rather considerable compared to the BASE cases. The biases in the LTA

NLDN cases are most favorable with values negative but closest to zero (small underestimate).

The LTA WWLLN cases produced larger negative biases than the BASE cases and LTA NLDN




cases, again, related to detection efficiency of the networks. All the LTA cases (both NLDN and

WWLLN) produced smaller errors than the BASE cases, and the differences between the NLDN

cases and WWLLN cases were minimal. Comparing the LTA cases with the BASE cases, one

noticeable feature is that with the different trigger and cudt values, all the statistics fluctuated

dramatically from one case to another in the BASE cases, but fluctuation among the LTA cases

was minimized and negligible. This is expected, as the moisture and temperature perturbations

used to trigger convection with LTA (Section 2) will take precedence over the trigger options

and grouping the lightning data into 30-minute bins should mitigate the influence of the cudt

option.  These features were deliberately incorporated into the LTA technique for precisely these

reasons, but this paper documents their systematic testing.

Examination of the statistics across the climatological regions over the CONUS domain

indicates that the Ohio Valley (OVC) stands out among all the regions with the lowest

correlation coefficients and largest RMSE values in all the BASE cases. However, with LTA, the

correlation coefficients in OVC were brought to the median range among other regions, though

the RMSE values were still the largest in that region; these features in OVC are more

understandable as manifested in Figure 12, examined in detail in Section 5. Other statistics in

OVC with LTA were comparable with other regions except for relatively larger negative MB

values associated with the LTA WWLLN cases. Another obvious characteristic with regards to

correlation coefficients and errors (RMSE and NME) was that there was more spread among the

regions in the LTA cases than in the BASE cases (except in OVC), which resulted from the

geographically heterogeneous nature of convective precipitation and the associated observed

lightning intensity across the regions.



To alleviate the underestimation of precipitation in the LTA WWLLN cases, additional
simulations (K1C10Ws0 and K2C10Ws0; where K1C10W and K2C10W are the same as in
Table 1, while s0 means zero suppress when lightning flash is not present) were performed by
switching the suppression option as described in Heath et al. (2016) from "ShallowOnly" to
"NoSuppress."  This modification still triggers deep convection where lightning is observed;
however, at grid points without lightning, the KF scheme is configured to run normally (i.e., the
same as in the BASE cases). As shown in Figure S1, the correlation coefficients in the
WWLLN+s0 cases were comparable with other LTA cases, and the values in the K2C10Ws0
case were similar to the NLDN cases and improved upon the K1C10W case. The MB in the
WWLLN+s0 cases were mostly positive (overestimate), which is expected because the KF
scheme has more freedom to activate deep convection. The K2C10Ws0 case produced the most
desirable results (domain mean MB is nearly zero) among all the cases. However, the biases
associated with LTA simulations using the "NoSuppress" option are affected by both the
lightning detection efficiency and the domain resolutions, which is more evident in the LTA
simulations over the hemispheric domain in Section 5.
4.2. Other Near-Surface Meteorological Variables
Besides precipitation, T2, water vapor mixing ratio, wind speed, and wind direction are
also analyzed. As shown in Figure 4, T2 in the BASE cases has correlation coefficients over the
CONUS domain and all the regions ranging from ~0.95–0.98. With LTA, the correlations for T2
were further improved for all the regions, with WWLLN cases performing slightly worse than
the NLDN cases. The impact of cumulus parameters on correlations was minimal for the BASE
and LTA cases. However, the cumulus parameters seem to impact the biases (MB and NMB,
Figures 4b,c) and errors (RMSE and NME, Figure 4d,e) in the BASE cases across all the regions,





and like precipitation, all the LTA cases minimized the impact of different cumulus parameter
values. All the LTA cases reduced the errors (RMSE and NME) associated with T2 across all the
regions, with NLDN slightly better than WWLLN. In summary, the T2 statistics were improved
by using LTA, and the WWLLN cases were comparable to the NLDN cases with a slight
degradation for all the regions.

The 2-m water vapor mixing ratios metrics (Figure 5) of the cases, in general, resemble

those of T2, in that the LTA cases have slightly increased the correlation coefficients from the
already well-simulated BASE cases. More spread occurs for biases (MB and NMB, Figures 5b,c)
and within the BASE cases for errors (RMSE and NME, Figures 5d,e). Regional spread in these
statistics is attributed to the diverse air mass types that drive large differences in the moisture
content and convective activity. Even though the values were low for both errors and biases (<
0.5%), using either LTA technique is an improvement over the BASE cases.

The cumulus parameters and LTA showed less impact on the correlations for 10-m wind

speed, but the impacts on biases and errors were noticeable. All the model cases underestimate
wind speed (~5–12%, depending on regions and model cases), and the cumulus parameters
caused relatively large differences in the metrics of the BASE cases with both trigger and cudt
options contributing most. Overall, using Trig2 with Cudt10 is most favorable in terms of biases
(less underestimate) and errors (smaller errors) among the BASE cases. In all the LTA cases, the
underestimation was reduced when compared to the BASE cases, and errors were reduced with
negligible differences among the cases with different cumulus parameters and assimilating
lightning data from the different networks. Similar behavior was observed for wind direction
where only correlation coefficient, MB, and RMSE are displayed in Figure S2 because
normalized metrics do not apply.






### 5. Northern Hemispheric WRF Simulations

As shown in Table 1, the model cases performed over the Northern Hemisphere are
similar to those performed over the CONUS, but with LTA cases using lightning data from
WWLLN that was gridded on the domain with 108-km horizontal grid spacing.
5.1. Precipitation
Before comparing the simulated precipitation with available observations, the examination
begins with how the WRF-simulated precipitation with and without LTA compares spatially over
the Northern Hemisphere. Figure 7 displays the mean daily precipitation during July 2016 from
two LTA cases and two BASE cases (Trig1 and Trig2) and the corresponding differences between
LTA and BASE (LTA – BASE) cases with the same trigger values, and Figure S3 presents the
mean daily precipitation differences between HK1C0W and HK1C0B cases throughout 2016.
Compared to the BASE cases, the LTA cases produced significantly less rainfall along the
Equatorial regions but generally more rainfall away from the Equator, especially over the
midlatitude land regions. Because no gauge-based observational data are available over the ocean,
the IMERG precipitation for July 2016 is presented in Figure 7g with the difference plots from the
base case (HK1C0B) and the LTA case (HK1C0W) being displayed in Figures 7h and 7i,
respectively. Over the Equatorial regions, the precipitation simulated by the LTA cases (Figures
7b and 7e) more closely resembled the IMERG precipitation than the BASE cases. The difference
plots clearly indicate that the base cases significantly overestimated, and the LTA cases slightly
underestimated the precipitation over large areas in the Equatorial regions. Similar results persisted
throughout the year as shown in Figure S4 (the difference of mean daily precipitation by month
between the base case, HK1C0B, and the IMERG product) and Figure S5 (the difference of mean



daily precipitation by month between the LTA case, HK1COW, and the IMERG product). Next,
the WRF simulated precipitation is compared with the CPC gauge-based analysis values over land.
Figure 8 displays the CPC rainfall and simulated mean daily precipitation during July 2016 along
with the estimates from the LTA and BASE cases with different cumulus parameters. Since the
gauge-based observational values are only available over land, the simulated values in Figure 8
are only displayed over land. As shown in Figure 8, all the model cases simulated the overall
spatial pattern of higher values in the tropical regions and lower values in high latitude regions.
However, subtle differences existed from case to case in different regions. For example, the
HK1C10B case (Figure 8d) and the HK2C10B case (Figure 8f) produced the highest and the lowest
precipitation over Africa and South America (along the Mexico coast to the South American
continent) within the modeling domain.

All the LTA cases uniformly produced larger correlation coefficients than the BASE

cases (Figure 9) when and where convective activities were prevalent. In the U.S., convective
activities occur during warm months (from May to September), while in Mexico and India,
convection is active throughout the year. In Canada, convective activities are less frequent
because of the cooler temperatures and low moisture at the high latitude. When and where
convection was active, the cumulus parameters produced significant differences in modeled
convective activity, as correlation coefficients are higher in the BASE cases with Trig1. Same as
the simulations over the CONUS domain, the cumulus parameters had a minor impact on the
correlation coefficients for the LTA cases regardless the regions. This indicates that, even with
the less dense WWLLN lightning observations, using LTA improves the timing and location of
deep convection.





RMSE were comparable for all the model cases across the selected regions (Figure 10),
with the LTA cases pointing to lower values than the BASE cases at all the regions except for the
U.S. where the LTA and BASE cases alternated to have slightly lower RMSE values over each
other during the year. Alternatively, the MB values varied significantly among the model cases
and across the regions as shown in Figure 11. One common feature is that the differences among
the LTA cases were small, but two distinctly separate groups among the BASE cases in all the
regions; the cases with Trig1 had always significantly greater precipitation values than the cases
with Trig2. In China and Mexico, all the simulations overestimated the precipitation through the
year except for small underestimate during the cool months (October–December). In India, the
overestimate and underestimate were equally split among the model cases, with dramatic
changes from month to month. The behavior of MB values among the model cases and through
the year was more stable for the U.S. (to a lesser extent in Canada) than in other regions, in
which the BASE cases with Trig1 have the best performance (MB values near zero), the BASE
cases with Trig2 significantly underestimated precipitation over land during convective season,
and all the LTA cases overestimated precipitation over land during the warm months. Here we
offer two plausible explanations for the drastically different behaviors of the MB values
associated with precipitation in different regions.
First, from the modeling point of view, the WRF model is widely studied and applied in
North America, especially in the U.S. As a result, more accurate observation-based datasets are
available to nudge WRF through FDDA (Liu et al., 2008), and all the work has led to the best
performance over the U.S. for the recommended default set of convective trigger and update
frequency for the cumulus scheme. Second, from the observational point of view, the CPC
rainfall dataset is built upon field gauge measurements that may vary in accuracy and





consistency from county to county. As shown in Figure S6, the NMB values were generally in
the range of -50% to 50% in the U.S. and Canada (comparable to the NMB values for the 12-km
CONUS simulations against PRISM precipitation as shown in Figure 3c), but in other countries,
especially during cool months, the values were up to hundreds or even thousands of percent that
suggests possible few observations available in the denominator in NMB calculations. For
instance, the highest NMB value in China coincided with the Spring Festival that is often a long
holiday for China suggesting possible gaps for data collection.

We next focus on the high MB values associated with the LTA cases in the U.S.

Consistent with the analysis in Figure 3b, the LTA WWLLN cases over the 12-km CONUS
domain always had larger negative MB (underestimates) than the LTA NLDN cases due to the
lower detection efficiency of lightning flashes in WWLLN than in NLDN. However, in the 108-
km hemispheric simulations, the same WWLLN datasets produced large positive MB
(overestimates) for precipitation. To understand this phenomenon, we need to first examine how
the LTA method works.  Because it uses a yes/no lightning indicator to trigger convection, 108-
km grid spacing might be too coarse for such a simplistic approach to work.  For example, one
lightning strike within a 108-km grid cell will trigger deep convection, which, because of the
large spatial coverage of the grid cell, can contribute to the high bias in precipitation because
convective rainfall is realistically more localized.  Although the KF scheme sets a fixed radius
for thunderstorms (e.g., Equation 6 in Kain 2004), applying the resulting rain over the entire 108-
km × 108-km grid box could partially explain the excess rainfall. This may also be explained by
the fact that the convective time-scale formulation in KF scheme was originally developed at
grid lengths of 20–25 km (Sims et al., 2017). A potential developmental pathway for the LTA
method at these scales is to test different thresholds of the 30-min flash density to ensure



sufficient lightning is present to trigger deep convection. Overall, compared to the CPC rainfall,
the LTA technique significantly improved the temporal and spatial correlation of convective
precipitation, but the precipitation amount was overestimated over the U.S. and other regions for
the 108-km modeling domain.

To further examine the impact of modeling domain resolutions on convective

precipitation, Figure 12 displays the spatial precipitation from PRISM, CPC (regridded onto the
12-km CONUS domain), and simulated precipitation from one BASE case and two LTA cases
with NLDN and WWLLN data, respectively, over the 12-km CONUS domain and one LTA case
over the 108-km hemispheric domain that has been regridded to the 12-km CONUS domain. As
shown in Figures 12a,b, the two observation-based precipitation products, PRISM and CPC,
compared well to each other, noting that the PRISM product displays more subtle granularity
than the CPC product due to the large difference in spatial resolutions (4-km for PRISM versus
0.5° for CPC).  The overall spatial pattern of mean daily precipitation was captured by both the
12-km LTA simulations (Figures 12d,e), and the 108-km LTA simulation (Figure 12f). The
heaviest rainfall was centered in the OVC area in the observation-based and the simulated
precipitation maps, but the shape and spread of the rain band were different. The rain band in the
12-km BASE case (Figure 12c) was more spread and scattered with southwest-to-northeast
orientation, while the observation-based products and the LTA cases indicated a relatively
smaller area with west-east direction. Thus, the LTA cases (12-km CONUS simulations)
compared better to the observation-based products spatially than the BASE case. The K2C10W
case (with WWLLN) tended to produce less precipitation than the K2C10N case (with NLDN)
and both observation-based products. These spatial discrepancies for precipitation in OVC
between PRISM and the model cases were reflected by the unique statistical behavior as





displayed in Figure 3 and discussed in Section 4.1. As a likely artifact of excessively activated
convection within the 108-km grid cells with a spatial scale much larger than most thunderstorm
scales, the HK2C10W case indicated areas of heavy precipitation that were also shown in the
observation-based products and the 12-km LTA cases at approximately same locations but with
much less spatial extent. To resonate with the large discrepancies in the MB values shown in
Figure 11a among the BASE cases, the precipitation from HK2C10B and HK2C10B cases is
similarly displayed in Figures 12g,h. The case with Trig1 was clearly more comparable to the
CONUS cases than the Trig2 case in that the precipitation was severely underestimated across
the entire U.S. These hemispheric simulations amplified the impact of the trigger options on
precipitation during warm months among the BASE cases, resulting in differences in daily total
precipitation of up to 40% in the U.S. (Figure S6a). These results underscore the need to
carefully set cumulus parameters for the KF scheme in WRF simulations.

The mismatch of the spatial scales between domain resolution and thunderstorms in the

108-km simulations is a limitation of current LTA scheme that could be improved in future
development. In addition to using lightning density to trigger convection, another option is to
implement the LTA scheme in the MultiScale Kain-Fritsch (MSKF) scheme (Glotfelty et al.,
2019; Zheng et al., 2016), a "scale-aware" variant of KF that refines the convective tendencies
based on the grid spacing used in the simulation.
5.2. Impact on Other Meteorological Variables

The impact of the cumulus parameters and LTA scheme on near-surface meteorological

variables of the 108-km hemispheric simulations are evaluated like the 12-km CONUS
simulations. However, due to the lack of observation data beyond North America, the analysis is
mainly focused on the U.S. regions, but all the available data within the hemispheric domain is





collectively referred to as "ALL" regardless of where the data originated. Affected by the coarser
domain resolution, all the statistical measures for T2 (Figure 13) from the hemispheric
simulations indicated degradations in model performance relative to the 12-km CONUS domain
(Figure 4). As in the CONUS simulations, the LTA cases increased correlation coefficients and
decreased errors (RMSE and NME) compared to the BASE cases. Like the CONUS simulations,
the cumulus parameters minimally affected the LTA cases, while significant deviations were
produced among the BASE cases. Unlike the CONUS simulations where both trigger and cudt
contributed to T2 differences, the large differences among the BASE cases for the hemispheric
simulations were attributed to the trigger options. Though all the cases tended to underestimate
T2 (contrary to the CONUS simulations where T2 was generally overestimated), among the
BASE cases, greater underestimates were associated with Trig1 than Trig2. The LTA cases
uniformly underestimated T2 consistent with the Trig1 BASE cases. The performance of
hemispheric simulations for 2-m water vapor mixing ratio (Figure 14) resembles T2 in the
comparison to the CONUS simulations (Figure 5), which produced smaller correlation
coefficients and larger errors and biases (mainly overestimates for both CONUS and hemispheric
simulations). Without exception, the LTA cases consistently performed better in terms of
correlation coefficients and errors than the BASE cases. However, different from other
meteorological variables, the MB and NMB associated with water vapor mixing ratio are
affected by both cumulus parameters (trigger and cudt) for all the model cases (both BASE cases
and LTA cases). The LTA cases with Trig1 performed better than the cases with Trig2, and with
the same trigger value, cudt=0 is preferable to cudt=10; however, for the BASE cases, it was the
opposite, though with smaller differences. At the 108-km grid spacing, the 10-m wind speed





(Figure S7) and wind direction (not shown) statistics were comparable among the cumulus
parameters and the application of LTA.

**6.   Discussion and Recommendations**

This study corroborated that the simple observation-based LTA scheme implemented in

Heath et al. (2016) improved WRF simulated precipitation and other near-surface meteorological
variables as evidenced by the simulations over multiple spatial scales and over a longer test
period. Testing on a 12-km CONUS domain using lightning flashes from WWLLN instead of
NLDN slightly reduced the correlation coefficients and locally increased errors due to the lower
detection efficiency of WWLLN. The update of the LTA technique reduced the underestimate of
precipitation that was often reported in the application of WRF simulations conducted over the
CONUS domain (U.S. EPA, 2019). Changing lightning flash data from NDLN to WWLLN
resulted in additional underestimate of precipitation due to fewer lightning flashes in WWLLN
than the NLDN dataset. However, when the WWLLN data was used in the hemispheric
simulations, the model performance for precipitation over the Equatorial regions was
significantly improved from significant overestimation in the base cases to slight
underestimation in the LTA cases, and the precipitation over land was generally overestimated
during the convective season for almost all the selected regions, especially over North America.

The application of LTA in the hemispheric simulations with a 108-km domain exposed a

shortcoming of this simple LTA scheme. When the model grid cell is substantially larger than
most thunderstorm scales (Murphy and Konrad II, 2005), over-triggering of convection within the
entire grid cell leads to overestimated precipitation. With the current LTA implementation and
the high lightning detection efficiency network, such as NLDN, the 12-km grid spacing is
suitable for LTA because thunderstorms often have a radial distance of 1–10 km. When lightning



data from low detection efficiency networks (such as WWLLN) are used over finer resolution
domains (≤12 km), the "NoSuppress" option with LTA could balance increasing precipitation
while maintaining reasonable levels of uncertainty in the other variables for a more holistic
model evaluation. The effect of domain resolution on precipitation simulation with LTA
portends further development and improvement of the LTA technique. Two potential
developmental directions are to alter values of lightning flash density to trigger deep convection
and/or to implement the LTA scheme in the MSKF scheme in WRF to adapt to different
simulation scales. Preliminary experimentation on the 108-km scale (not shown) suggests that
MSKF could improve these comparisons with observations (compared to the KF scheme
presented here), including better cloud and precipitation fields (Hogrefe et al., 2021).
The experiment of cumulus parameters (trigger and cudt) associated with the KF scheme was
performed for both the CONUS and hemispheric WRF simulations. Results revealed several key
behaviors in both the BASE case simulations and LTA case simulations. First, the BASE case
simulations were sensitive to both trigger and cudt options over the CONUS domain, but only
trigger options produced significant variations for the hemispheric simulations. Second, the
impact of the cumulus parameters on LTA cases was insignificant for both modeling domains.
Separately, the original LTA technique as described in Heath et al. (2016) showed influence
from the cumulus parameters on the LTA cases (Figure S8), but after implementing the updates
described in Section 2, the fluctuations among the LTA cases were significantly reduced. Third,
the most pronounced impact of cumulus parameters was on the amount of precipitation in the
BASE cases. The Trig1 option generated up to a 10% overestimate of month-mean daily
precipitation over the CONUS with cudt=0 and an additional 10–15% overestimate with cudt=10
during July 2016.  With Trig2, the simulated precipitation became underestimated by about 10–





15%, with the cudt contributing to ~5% difference; Cudt10 had less underestimate than Cudt0.
However, over the hemispheric domain, only the trigger option dramatically affected simulated
precipitation; during the summer months (June, July, and August), the Trig2 cases
underestimated the mean daily precipitation by up to 40% more than the Trig1 cases that
matched the observation-based precipitation products within 10%. In summary, without LTA,
the recommended default values (trigger=1 and cudt=0) by WRF documentation remain the best
option for both the CONUS and hemispheric simulations to achieve the best model performance,
especially for North America, and with LTA, all the options performed equally well.

As one of the most prominent meteorological models, WRF has been widely used in a variety

of applications from regional to global scales and from weather and climate studies to air
pollution transport in air quality forecast and regulatory compliances. It is important to improve
the convective processes to have more accurate precipitation and other meteorological fields with
more resources being available including observational datasets, computing capability, and
advanced scheme development. Observation-based data assimilation has been historically proven
to be one of the most effective methods to improve model's performance in time and space. This
research is emerging to consider and use the lightning observations that have become available in
various formats and scales in the past decades to improve convection simulations through LTA.
Additional networks of lightning observations and more detailed properties associated with the
process of lightning discharge are becoming available (such as the scope and strength of
lightning energy level and the separation of cloud-to-ground and inter- or intra-cloud strikes
being more accurately quantified, especially with the available satellite lightning products from
Geostationary Lightning Mapper (GLM) detection systems borne on the GOES-16 and -17



satellites (Goodman et al., 2013)). Accordingly, lightning assimilation techniques will continue
to evolve and build upon the research presented here.



**Code and data availability**

The WRF model is available for download through the WRF website (http://www.wrf-model.org/index.php). The LTA code is not publicly available yet but interested users can contact the corresponding author to acquire the source code. The raw lightning flash observation data can be purchased through Vaisala Inc. (https://www.vaisala.com/en/products/systems/lightning-detection), and the WWLLN raw data are also available for purchase at http://wwlln.net. The immediate data except the lightning flash data behind the figures are available from doi: https://doi.org/10.5281/zenodo.6493145.

**Author contributions**. DK conceptualized the study, performed the model simulation and data curation, carried out the analysis, and wrote the paper. NH developed the mechanism and software and wrote the paper. RG prepared the scripts for model simulations and data analysis and edited the paper. TS supervised the research, provided resources, and edited the paper. JP edited the paper.

**Competing interests**. The authors declare that they have no conflict of interest.

**Disclaimer:** This paper has been subjected to an EPA review and approved for publication. The views expressed here are those of the authors and do not necessarily reflect the views or policies of the U.S. Environmental Protection Agency (EPA).

**Acknowledgement**:

We thank Jerry Herwehe and Kiran Alapaty at the EPA for reviewing the paper and providing valuable comments and suggestions. PRISM Precipitation data for the United States are retrieved from https://climatedataguide.ucar.edu/climate-data/prism-high-resolution-spatial-climate-data-united-states-maxmin-temp-dewpoint and the CPC Global Unified Precipitation data provided by the NOAA/OAR/ESRL PSL, Boulder, Colorado, USA, from their Web site at https://psl.noaa.gov/data/gridded/data.cpc.globalprecip.html. The IMERG data were provided by the NASA/Goddard Space Flight Center's Precipitation Measurement Missions (PMM) Science Team and Precipitation Processing System (PPS), which develop and compute the IMERG as a contribution to GPM, and archived at the NASA GES DISC.



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




**Table 1. Model Cases (N/A: Not Applicable)**


| Case Name | trigger (K1 or K2) | cudt (C0 or C10) | LTA (B, N, W) | Network | Domain |
|-----------|--------------------|------------------|---------------|---------|--------|
| K1C0B | 1 | 0 | NO | N/A | CONUS |
| K1C10B | 1 | 10 | NO | N/A | CONUS |
| K2C0B | 2 | 0 | NO | N/A | CONUS |
| K2C10B | 2 | 10 | NO | N/A | CONUS |
| K1C0N | 1 | 0 | YES | NLDN | CONUS |
| K1C10N | 1 | 10 | YES | NLDN | CONUS |
| K2C0N | 2 | 0 | YES | NLDN | CONUS |
| K2C10N | 2 | 10 | YES | NLDN | CONUS |
| K1C0W | 1 | 0 | YES | WWLLN | CONUS |
| K1C10W | 1 | 10 | YES | WWLLN | CONUS |
| K2C0W | 2 | 0 | YES | WWLLN | CONUS |
| K2C10W | 2 | 10 | YES | WWLLN | CONUS |
| HK1C0B | 1 | 0 | NO | N/A | Hemisphere |
| HK1C10B | 1 | 10 | NO | N/A | Hemisphere |
| HK2C0B | 2 | 0 | NO | N/A | Hemisphere |
| HK2C10B | 2 | 10 | NO | N/A | Hemisphere |
| HK1C0W | 1 | 0 | YES | WWLLN | Hemisphere |
| HK1C10W | 1 | 10 | YES | WWLLN | Hemisphere |
| HK2C0W | 2 | 0 | YES | WWLLN | Hemisphere |
| HK2C10W | 2 | 10 | YES | WWLLN | Hemisphere |









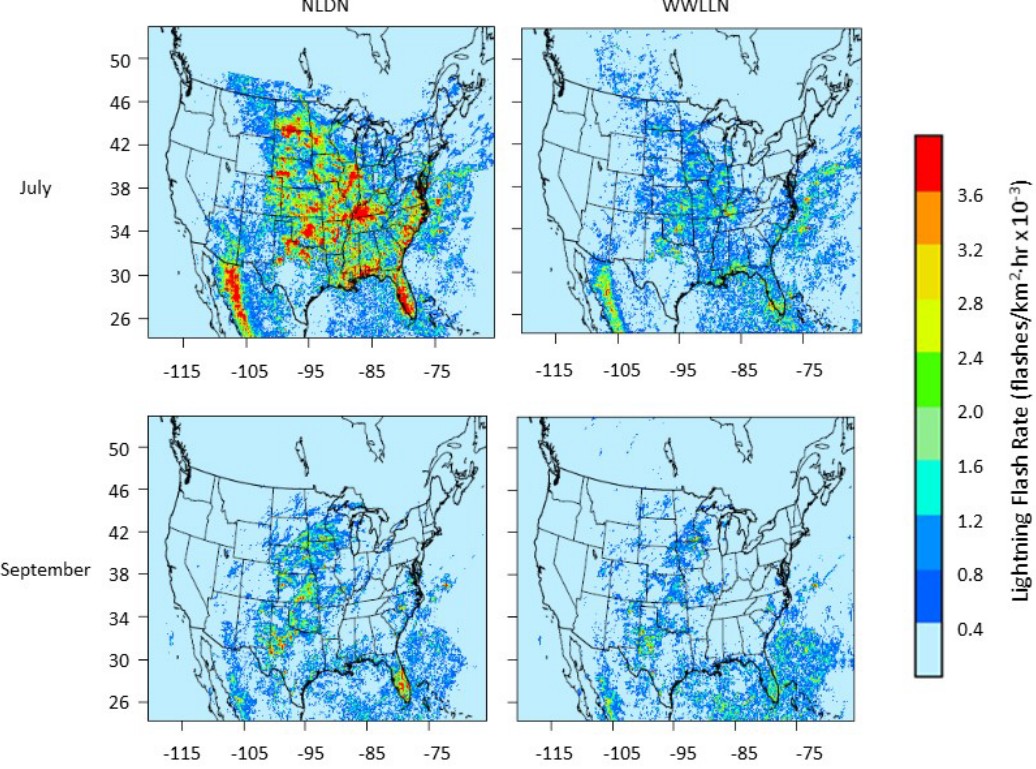


**Figure 1.** The mean hourly lightning flash rate from NLDN and WWLLN over the 12km

CONUS domain in July and September 2016.









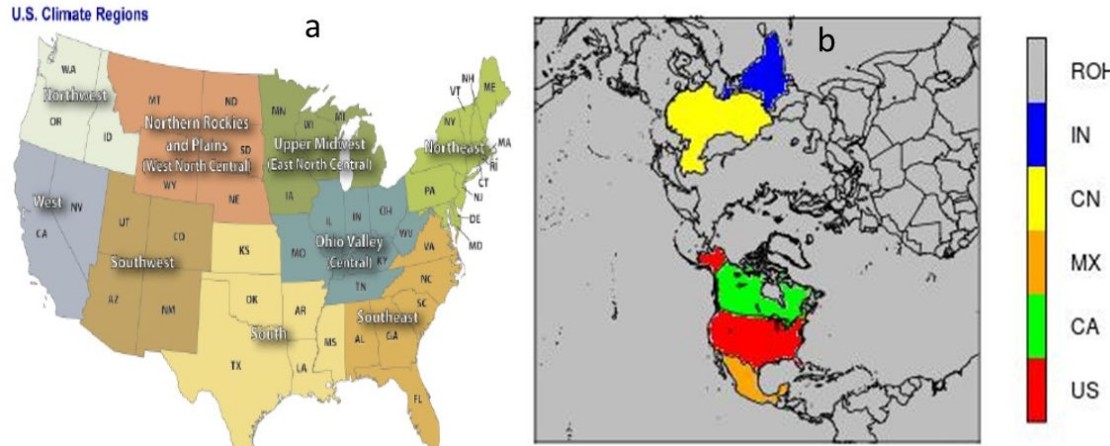

**Figure 2.** Analysis Regions (Countries), a. the climate regions in the CONUS, and b. the

countries over the northern hemisphere – US: United States; CA: Canada; MX: Mexico; CN:

China; IN: India; ROH: Other countries/regions except the five specific countries in the

hemispheric domain. The U.S. climate regions are: Northeast (NE), Southeast (SE), Ohio Valey

Central (OVC), Upper Midwest (UM), South, West North Central (WNC), Southwest (SW),

Northwest (NW), and West.





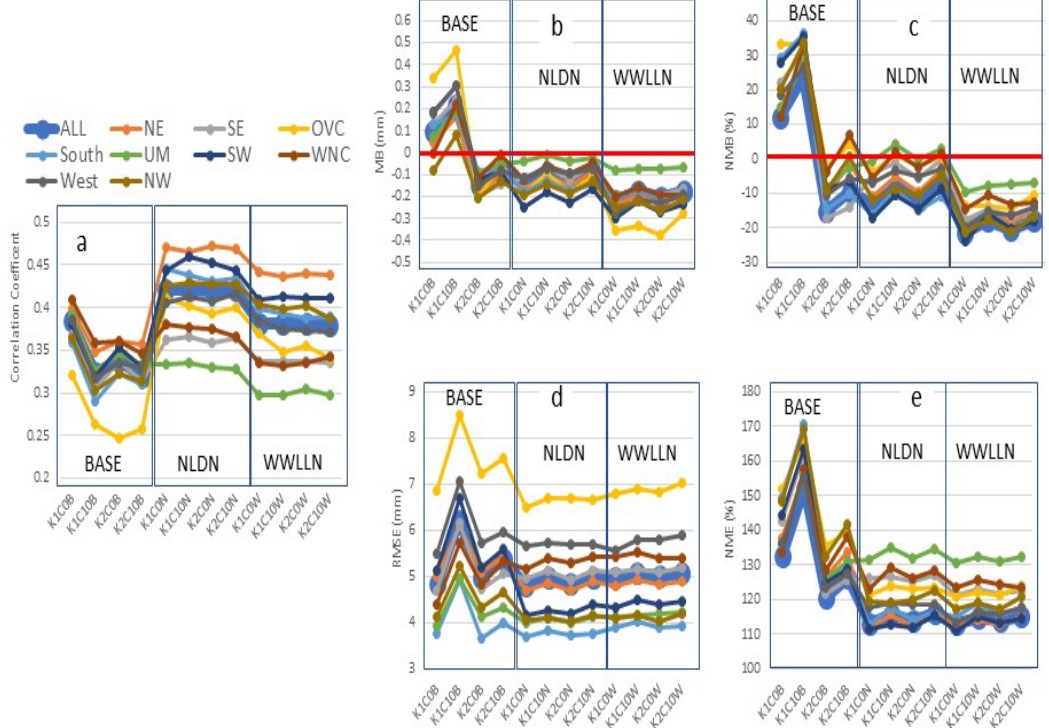

798

**Figure 3.** Monthly mean statistics for precipitation from BASE and LTA simulations

comparing to the values from PRISM for the modeling domain and the climatological

regions over the CONUS, respectively, during July 2016: a) correlation coefficient, b) MB,

c) NMB, d) RMSE, and e) NME. In each plot, there are three sets of simulations (BASE,

LTA with NLDN, and LTA with WWLLN) and each having four cases from the

combinations of cumulus parameters.





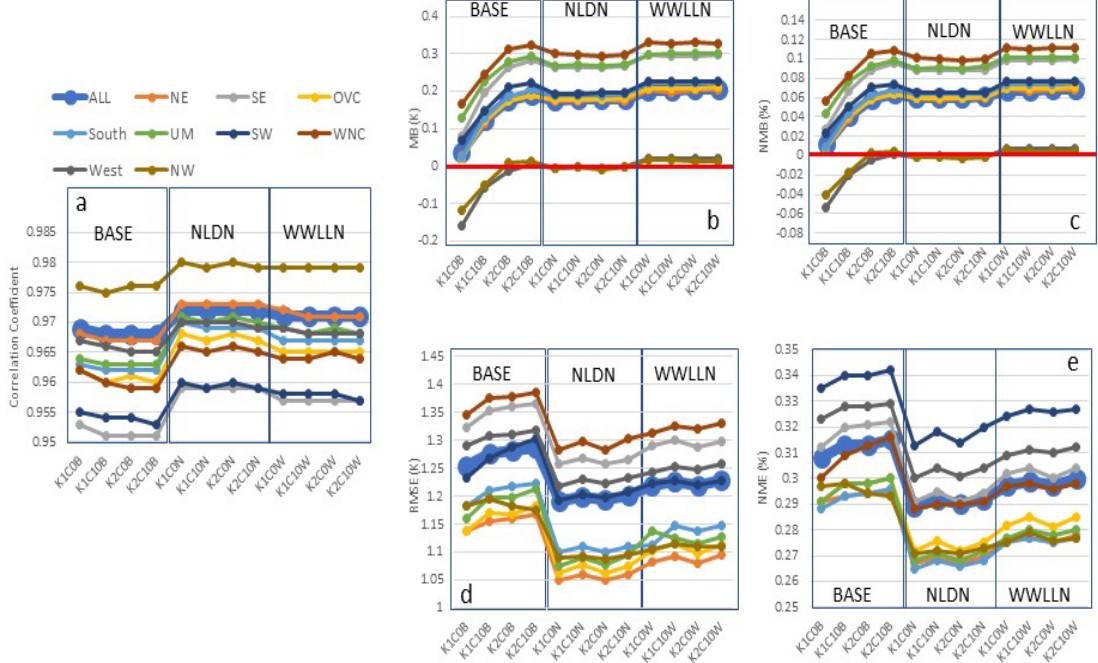


**Figure 4.** Same as Figure 3, but for 2-m temperature (T2) in that the simulated T2 values are
paired with observations from NCEI's land-based stations in time and space (hourly mean
values).





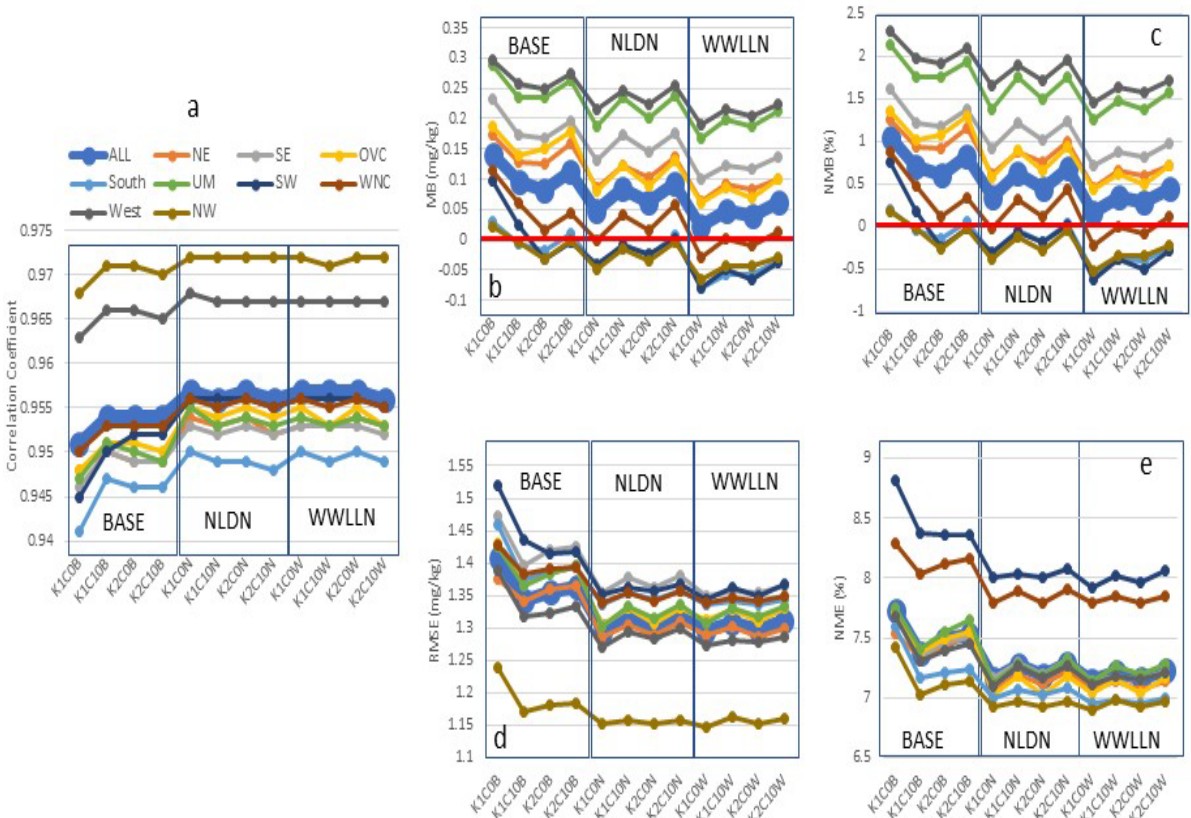


**Figure 5.** Same as Figure 4, but for 2-m water vapor mixing ratio.





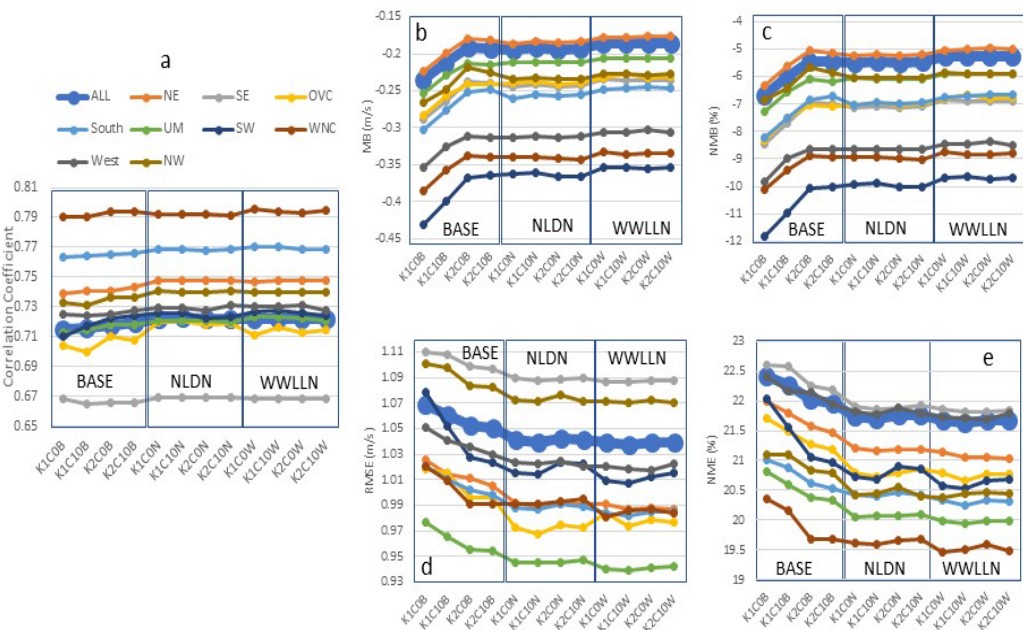

**Figure 6.** Same as Figure 4, but for 10-m wind speed.



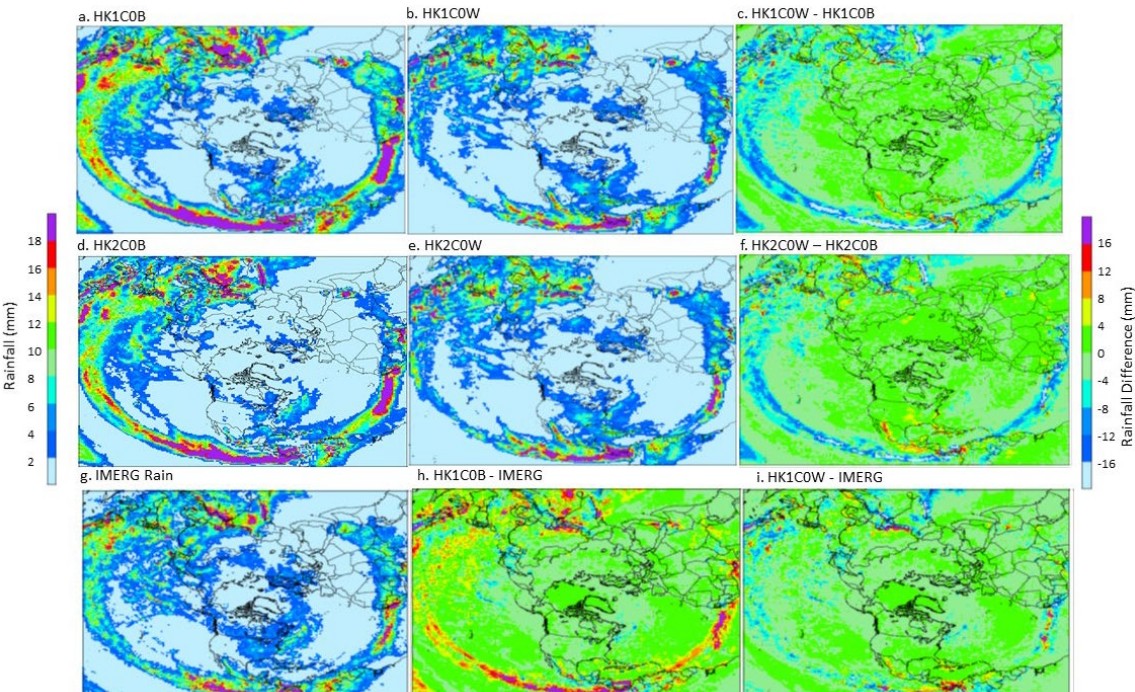

**Figure 7**. The mean daily rainfall during July 2016 simulated by base model cases (a. HK1C0B
and d. HK2C0B), LTA cases (b. HK1C0W and e. HK2C0W), and the satellite GPM
produced rainfall (g), and the differences between the LTA and BASE cases (c.
HK1C0W – HK1C0B and f. HK2C0W – HK2C0B) and between the simulated cases and
satellite IMERG products (h. HK1C0B – IMERG and i. HK1C0W – IMERG). Note that
the left legend applies to the rain maps (a, b, d, e, and g), and the right legend applies to
the difference plots (c, f, h, and i).



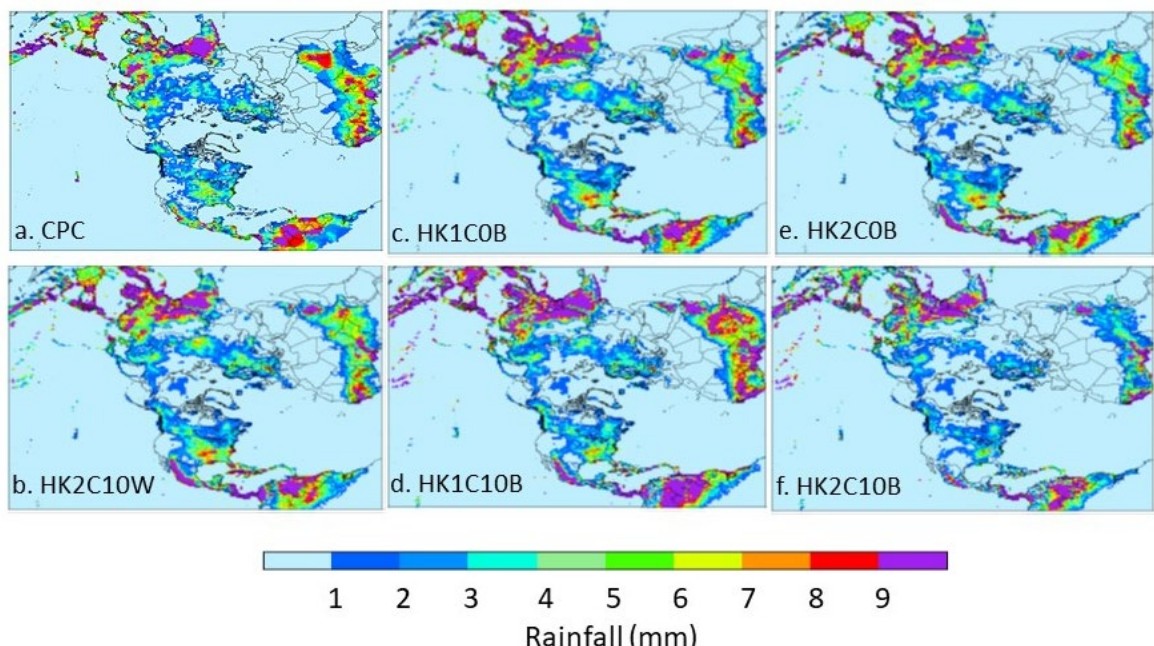

823

**Figure 8.** CPC rainfall (a) and simulated (b-f) mean daily precipitation during July 2016 over the
hemispheric domain. The LTA configuration is represented by one case (b. HK2C10W) since all
the LTA cases with different cumulus parameters produced similar results. All BASE cases are
shown here (c-f) because the cumulus parameters do impact the simulated precipitation when not
using LTA.





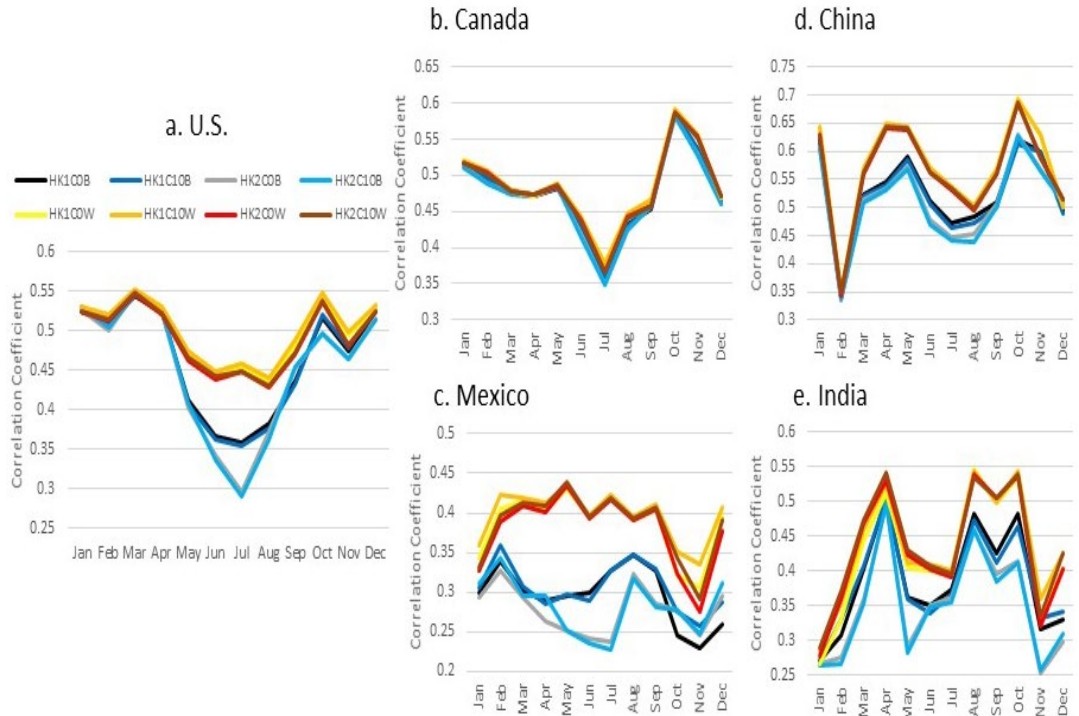


**Figure 9.** The monthly correlation coefficient between CPC and simulated precipitation in
selected countries: a. United States, b. Canada, c. Mexico, d, China, and e. India. Note
that all the BASE cases are plotted in cool colors and LTA cases in warm colors.



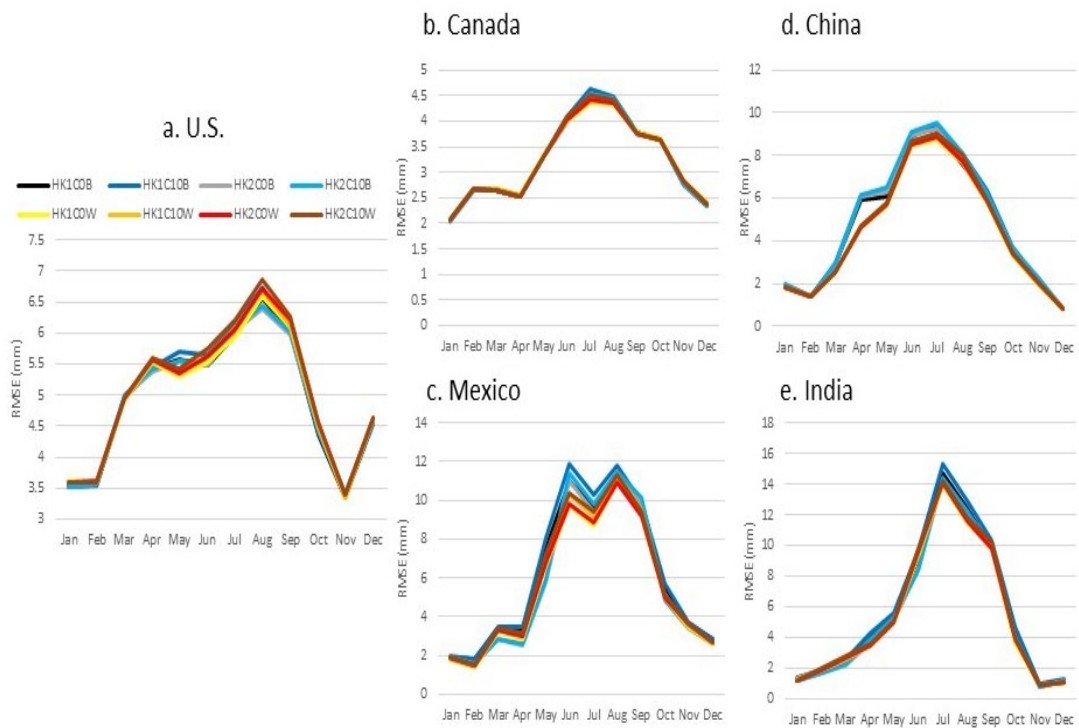


**Figure 10.** Same as Figure 8, but for RMSE.





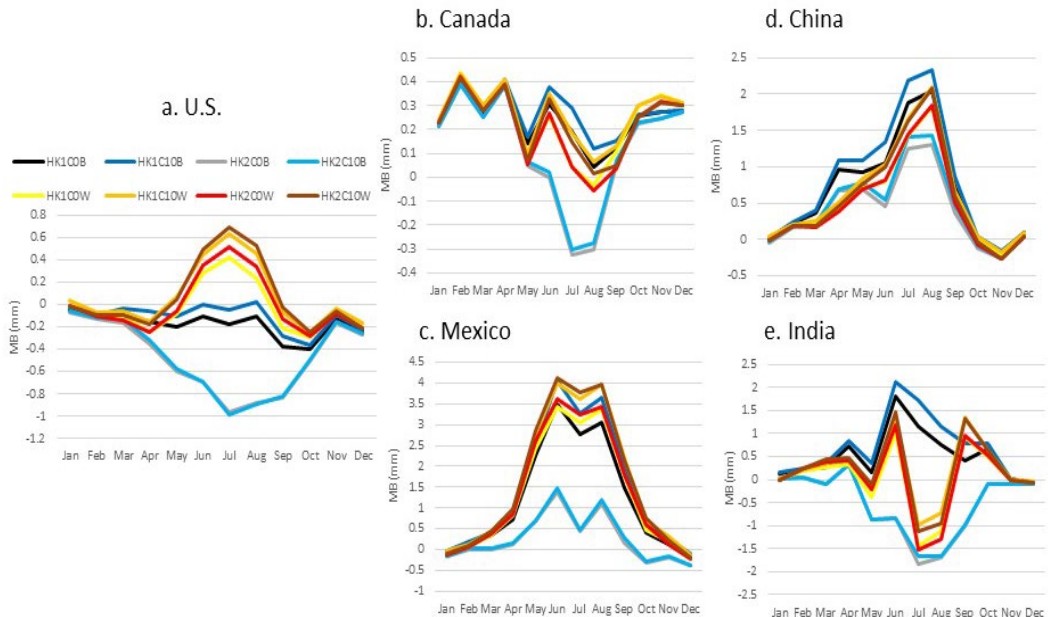


**Figure 11.** Same as Figure 8, but for MB.



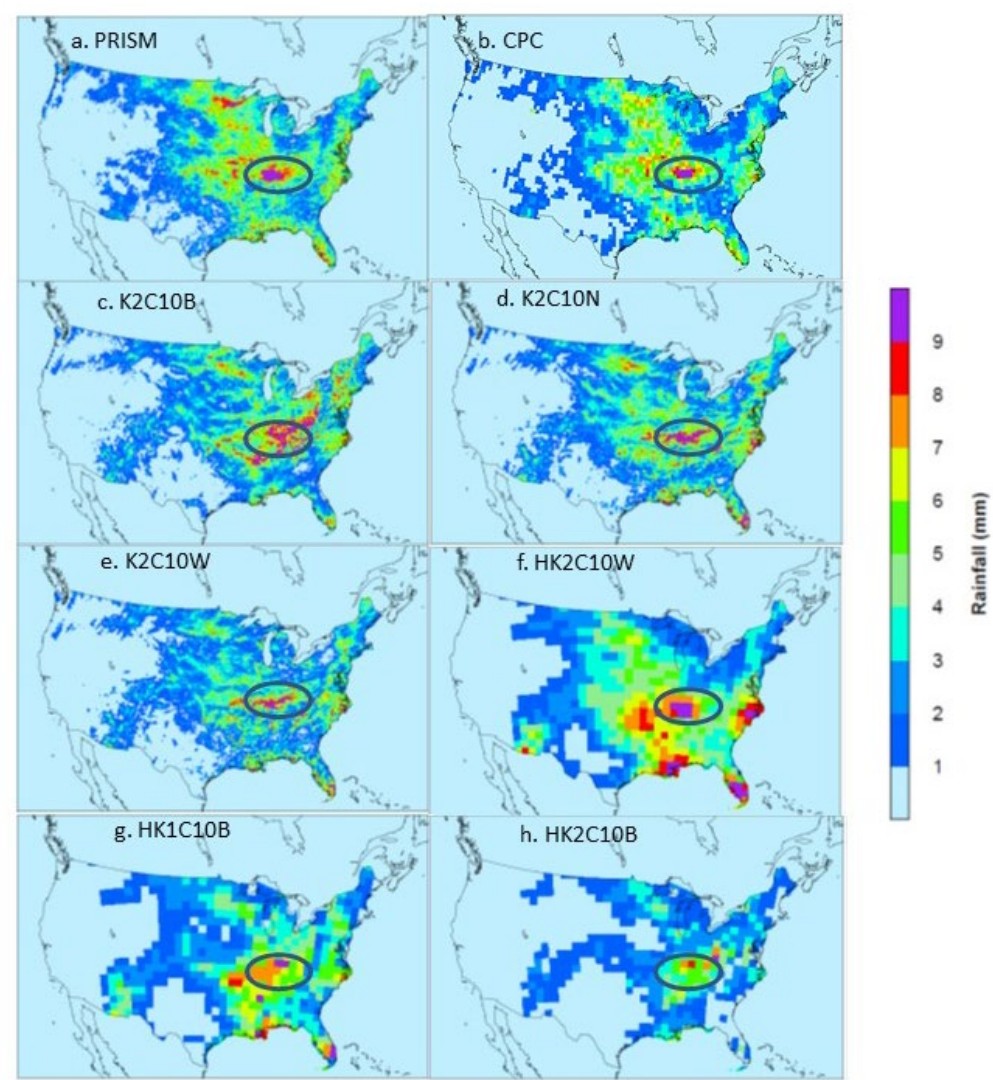


**Figure 12.** Mean daily precipitation over the CONUS during July 2016 from a) PRISM, b) CPC,
c) K2C10B, d) K2C10N, e) K2C10W, and f) HK2C10W, g) HK1C10B, and h)
HK2C10B. Note that all the observational based products and the 108 km hemispheric
simulations are regridded onto the 12 km CONUS domain.






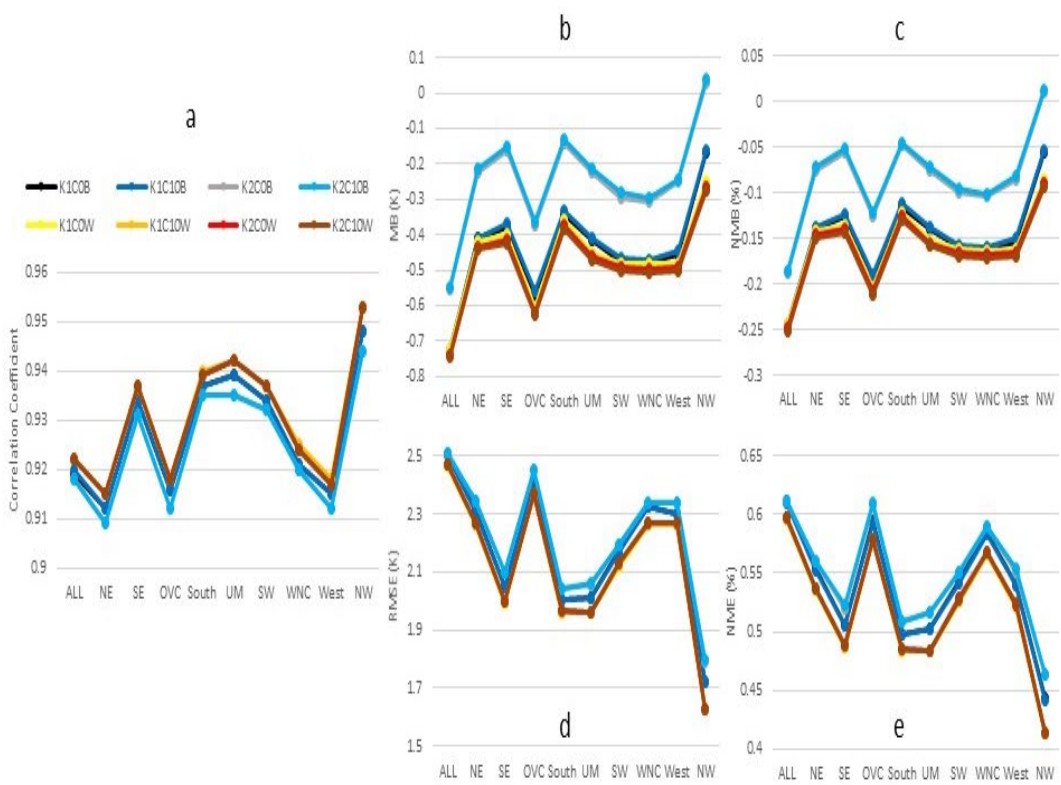


**Figure 13.** Monthly mean statistics for 2-m temperature from hemispheric BASE and LTA
simulations comparing to surface observations during July 2016: a) correlation coefficient, b)
MB, c) NMB, d) RMSE, and e) NME.





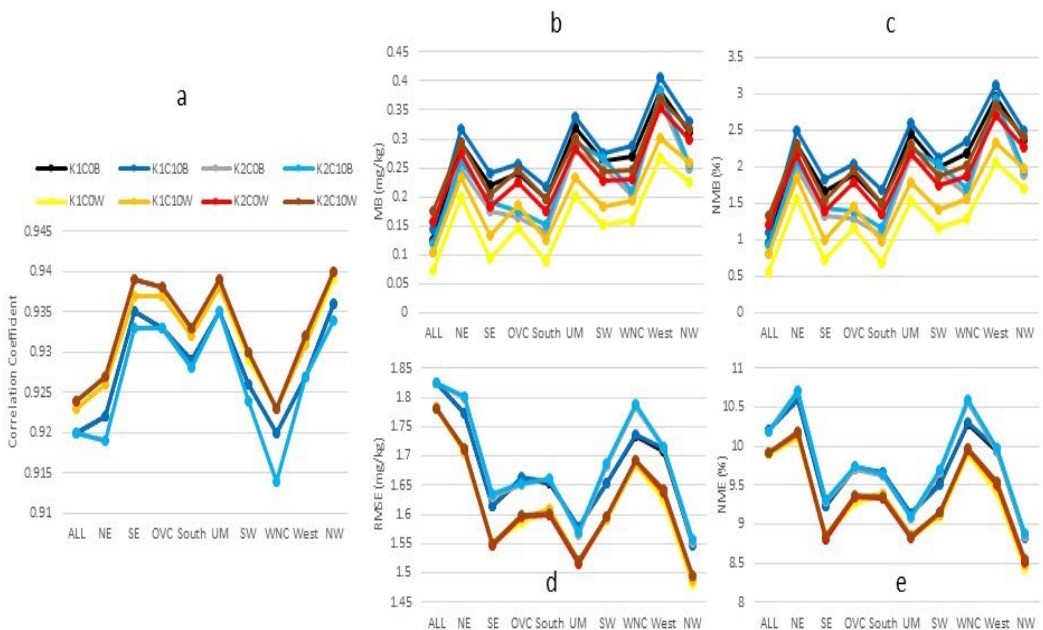


**Figure 14.** Same as Figure 12, but for 2-m water vapor mixing ratio.
