# Peer review of "Lightning Assimilation in the WRF-model (Version 4.1.1): Technique Updates and Assessment of the Applications from Regional to Hemispheric Scales Daiwen Kang1\*, Nicholas K. Heath2#, Robert C. Gilliam1, Tanya L. Spero1, and Jonathan E. Pleim1</sup"

_EGUsphere, 2022_

## Author Response (AR1)

**Responses to Reviewer 1:**

** General comments

The submitted manuscript describes how a method for convection in the WRF model to observed lightning changes its sensitivity to various namelist parameters, at both local and regional spatial scales. The paper is informative and very well organized, but heavy on details that may not serve the overall message (more on this below). I recommend it for publication following minor revisions, which are outlined below.

**We thank Dr. Neef for the overall positive assessment of this manuscript and appreciate the constructive comments and suggestions.**

** Specific comments

1. I am concerned about the use of the term "data assimilation" to describe the approach used here, which essentially uses the presence of observed lightning to trigger convection in the model, whereas the term "data assimilation" typically refers to complex systems that use some sort of variational or Kalman-filter type method to periodically update prognostic model variables with observations. "Lightning-triggered convection" might be a more apt descriptor, but I leave this up to the authors and editor to decide.

**Although some researchers consider "data assimilation" as you describe it, the term has been used more broadly to encompass other methods of observational influence throughout a simulation. Similarly, we previously used the term "lightning assimilation" in the Heath et al. (2016), and it has been adopted by the WRF community. To be consistent with existing terminology, we continued to use "lightning assimilation" here.**

2. The main thrust of the paper outlines how LTA changes the sensitivity of the convection to various namelist parameters in WRF. It's unclear, however, to what extent this result is interesting to users of other models. It looks like there are two major conclusions about the parameters (reading from lines 579-584 here): (1) that regional simulations are sensitive to both a parameters but hemispheric simulations only to the trigger parameter, and (2) that sensitivity to both of these parameters really goes down once LTA is turned on. I can see how the second of these is interesting beyond WRF (more constraint to data means less sensitivity to parameters) but the first is less intuitive. Can you zoom out and draw a more general, non-WRF-specific, conclusion?

**Although broader viewpoints would be useful outside the WRF model, the parameters tested in this study (kfeta_trigger and cudt) are specific to the Kain-Fritsch convective scheme. Different convective schemes within WRF have different limits in terms of the temporal and spatial scales and the application conditions. Without testing in another study, unfortunately, we cannot draw more generalized conclusions, even for other schemes within the WRF model.**

3. Lines 95-117: The introduction explains previous successes with LTA and the plan for the current paper, but without explaining what LTA actually is. As stated above, calling it "Lightning Data Assimilation" might imply something different than what is actually done (i.e. the assimilation of lightning data as part of the existing WRF 4D data assimilation), which makes it difficult to see why LTA should be tested in conjunction with two namelist parameters for convection. Paragraph 1 of section 2 has a great, concise summary of what LTA does -- I suggest moving this statement to the introduction.

**As you suggested, we revised the manuscript by moving part of paragraph 1 from Section 2 to the Introduction. In addition, we have rearranged and revised the descriptions related to the LTA technique in the introduction by removing redundant statements and improving the readability. The description of LTA technique has now changed to:**

> **"Heath et al. (2016) implemented an LTA technique in the Kain-Fritsch (KF) convective scheme (Kain, 2004) in the Weather Research and Forecasting (WRF) model, which extended the works of Rogers et al. (2000), Mansell et al. (2007), Lagouvardos et al. (2013), and Giannaros et al. (2016). In general, the lightning assimilation approach is straightforward, activating deep convection where lightning is observed and only allowing shallow convection where it is not. It was tested using WRFv3.8 simulations for several months in 2011 using lightning observations from the National Lightning Detection Network (NLDN) over the contiguous United States (CONUS). It was found that the simulation of warm-season rainfall was substantially improved, and other near-surface meteorological variables were clearly improved in retrospective WRF applications."**

4. Line 257: How many minutes is the model timestep? (Is cudt=10 more or less frequent?)

**The model timesteps for the CONUS and hemispheric simulations are 1 minute and 3 minutes, respectively, as indicated by "time_step" in the supplementary information (Tables S1 and S2). In response to your comment, we revised the text**

**by adding, "The time step is 1 minute (Table S1) and 3 minutes (Table S2) for the CONUS and hemispheric WRF simulations, respectively".**

5. The abbreviations for the experiment matrix (Table 1) are confusing! For example, since the names of individual WRF namelist parameters are pretty meaningless to people who don't use WRF, there's nothing intuitive about abbreviating which convective trigger is used as "K" or abbreviating the timing of the convective scheme as "C". I can't say what a more effective naming scheme would look like, but I found the abbreviations hard to remember, which made all the subsequent figures hard to understand. For example, the discussion of Figure 1 talks about the effects of trig 1 vs trig2, but in order to understand what is being talked about, the reader needs to connect the xaxis labels in 5 panels to entries in table 1 and figure out what all that means, and all that work is going to muddle the main results of the figure.

**We empathize with the reviewer (and the readers) in distinguishing these 20 model cases with two domains, two pairs of configuration sensitivities, and three options for lightning data from two networks, which is how we developed Table 1. We could not come up with a more succinct and intuitive naming convention, which means readers will need to refer to Table 1 to understand the related figures and discussions. In response to your comment, we have augmented the caption for Table 1 to describe the naming convention. We also combined columns that were previously labeled "LTA" and "Network", as they were redundant, and we rearranged the columns of Table 1 to align with the convention we used.**

6. Line 294: the "ShallowOnly" option is mentioned but not explained. It would be helpful, in Section 3 where the parameters of the convective scheme are described, to briefly explain how the trigger options change then LTA is on/off.

**Thank you for pointing this out. The "ShallowOnly" option was previously explained, but we did not actually specify that we were describing it. In the updated manuscript, the description of the ShallowOnly technique is in lines 174-178, and we added the following parenthetical: "this is referred to as the "ShallowOnly" method".**

7. Figure 3: I advise against the use of a line plot here, because connecting the dots with lines implies to the reader that they are have a temporal ordering, when really you are showing independent experiments. How about bar graphs? This would clearly show one of the major results of this figure, which is that the experiments with LTA off are much more sensitive to the choice of convection parameters. Also, it really necessary to show 5 different statistical measures? They seem to mostly reflect the same things

(correlation goes down, while bias and errors go up, when agreement with obs is poor) so it seems like there is a lot of redundant information in this figure.

**We understand your concern and respect your opinion. We tried different types of plots to represent the data. However, with many model cases across the climatological regions, we found that the line plots were most straightforward to interpret. As an example, the bar graph (below) is too busy to see the same region from different model cases, while the line plot presents the same information more clearly. We elected to use multiple statistical metrics to be comprehensive in our analysis.**

[Figure]

[Figure]

8. Figure 3: It's unclear from the text _why_ LTA has the strongest effect on the Ohio Valley -- I assume it would be because there is a lot of lightning in this region in the summer, but the same is true for the Southeast and the Upper Midwest, and those regions have a much smaller effect on LTA. Is there a simple explanation?

**The varying effect of LTA at different regions may be attributed to the differing performance of the base model and the differing relationship between convective precipitation and lightning strikes (Kang et al., 2019). As Figures 3 and 4 indicated in Kang et al. (2019)—shown below—the relationship between convective precipitation and lightning strikes varies across the country.**

[Figure]

[Figure]

*Figure 3. (a) The ratio (background) between lightning flash density and modeled convective precipitation (CP) in July (2002–2014; similar patterns for other months are not shown) and the analysis regions (R1 to R5). (b) Comparison of monthly mean NLDN lightning flash density (km$^{-2}$ h$^{-1}$) and modeled convective precipitation for the domain (All) and regions (R1 to R5) from 2002 to 2014. Each plotted pixel represents the monthly mean value: 13 (years)x12 (months) total pixels over each region. [From Kang et al. (2019)]*

[Figure]

*Figure 4. Comparison of monthly mean NLDN lightning flash density (km$^{-2}$ h$^{-1}$) and modeled convective precipitation for the west (green, region 1 from Fig. 3a) and east (blue, regions 2–5 in Fig. 3a) from 2002–2014 on a (a) linear scale and (b) logarithmic scale. Each plotted pixel represents the monthly mean value: 13 (years)x12 (months) total pixels over each region. [From Kang et al. (2019)]*

9. Figs. 4-5: These figures show a lot of the same things as Fig 3, with the most noticeable difference being (I think?) that the different namelist parameters can really affect the bias in surface variables. However, it's hard to discern a real memorable message from this part -- Section 4.2 lists a lot of detailed results but doesn't really put them into a greater context. It would be great if these figures could be distilled down to the most salient results (perhaps by only showing one type of bias?) and then given more physical explanation rather than listing out the various details of each figure.

**We believe that Figs. 4 and 5 should be like Fig. 3 because we want to be comprehensive in illustrating the sensitivities on precipitation, 2-m temperature, and 2-m water vapor mixing ratio. We believe each of these metrics provides valuable context in interpreting our results, but it has been a long-standing debate, as noted by Hodson (2022). For example, some researchers prefer one set of metrics to the other (e.g., Chai and Draxler, 2014; Willmott and Matsuura,**

**2005), arguing that individual metrics are unique, which supports providing a suite of metrics. We believe that this suite of metrics provides transparency and allows interpretation in the appropriate contexts across the most relevant near-surface thermodynamic variables.**

10. Fig. 7: Column and row headings would be great here; without them, it's really hard to know which panel shows what (again, the experiment abbreviations aren't really intuitive). For the panels showing differences in rainfall, a divergent colormap (where zero is white) would be helpful (I realize that this can be pain to implement in a small-multiples plots so this is just a tip). More importantly, the only result discussed for Figure 7 is that LTA brings the simulated precipitation closer to the observations. If that's all this figure shows, are 9 panels really necessary? Again, it's a lot of information to confront the reader with, when the actual take home message of the figure is probably more simple.

**Thank you for your comments and suggestion. In response to your comment, we rearranged Figure 7 and added column headings. Figure 7 is a comparison to the IMERG dataset to illustrate how the LTA technique impacted the precipitation over the ocean, especially in the tropical regions where no ground-based observations are available.**

11. I'm not sure how Figure 8 fits into the larger premise of the paper. The main result seems to be that the different parameters produce subtle differences in the daily precipitation. Since you don't go into detail about these differences, it's maybe not necessary to include this figure. (Figure 9 seems far more informative and maybe covers your bases sufficiently?)

**While Figure 8 and Figure 9 present the comparison between simulated precipitation and the gauge-based observations which are only available over land. Figure 8 displays the spatial distribution of observed and simulated values, and Figure 9 presents the statistics at selected regions of interest. We think Figure 8 is important to retain, even if there are only subtle spatial differences.**

12. Line 450: FDDA is mentioned for the first time here; are all the simulations described here run with data assimilation? If yes, that should be made clear much earlier, i.e. when the experiments are described. Also, since most of the runs have significant biases in precipitation, it would be good to know roughly what types of data were assimilated in the FDDA framework.

**We updated the manuscript in section 3.4 (WRF configuration) to state how the FDDA was configured. In this study, we used the "analysis nudging" (or "grid nudging") in all WRF simulations. The hemispheric simulations were constrained**

**toward analyses from the NCEP Global Forecast System, while the CONUS simulations were constrained toward analyses from the NCEP North American Mesoscale (NAM) model. Nudging was applied in WRF model layers above the planetary boundary layer (dynamic in space and time). Nudging was applied for temperature, water vapor mixing ratio, and horizontal wind components with specifications following Heath et al. (2016).**

13. Figure 11: If I understood correctly, the authors are arguing that mean bias changes more drastically with different parameters settings for regions that are less constrained by observations and/or to which the model is less well tuned than the United States. But does this really bear out? Looking at the figure, I see MB for the USA vacillate wildly between the different experiments, even flipping sign. Also, why does MB change so drastically between experiments but RMSE doesn't? Since you show both, it would be good tp explain what each of these measures reflects.

**The bias changes more drastically from month to month from the changes within the same model case (i.e., the same parameter settings). For the USA, the vacillation is caused by the different convective parameters (primarily the convective trigger), but within the same model case, the month-to-month variations are mainly attributed to the intensity of convective events. By contrast, in India, the MB in the same model case changed from ~2.5 mm in June to -1.5 mm in July for the LTA cases. In response to your comment, we revised the text as, "with dramatic changes from month to month in the same model case".**

**MB (Fig. 11) changes more dramatically than RMSE (Fig. 10) because of differences in what is represented by these metrics. MB measures the deviation of simulated values from observations, while RMSE computes the root mean square of the difference. Between the different model cases, the MB vacillates from positive to negative with similar magnitude, while only the magnitude matters with the RMSE (regardless the sign). RMSE also amplifies large deviations. The differences between Figs. 10 and 11 support our use of different statistical metrics to reflect the different aspects of model performance. Since these statistical metrics are widely used in the atmospheric modeling community, we believe it is unnecessary to provide detailed description here.**

14. Line 468: On the hemispheric scale, LTA means triggering convection over a huge area by something as local and small scale as lightning -- it actually seems quite surprising that you have any success at all with LTA at these scales. It would be helpful to mention much earlier on what a stretch it is to try to apply LTA at these scales, and then emphasize what about it works (if I understand correctly, it's that LTA improves the correlation to rain gauge data, i.e. Fig. 9?).

**Thank you for the suggestion. However, we think that the current discussion flows well when we directly relate the results to the discussion. As a result, we decided to keep it as it was originally written.**

** Technical corrections

1. Line 103: Does the lower cost of WWLLN data mean a lower cost to users (i.e. it costs less to obtain the data) or a lower cost of the actual measurements? Is there a simple reason why this is the case?

**The lower cost is both ways: lower cost to the users and a lower cost of the actual measurements compared to the NLDN data. Since this information is not critical, we removed the statement in the revised manuscript.**

2. Line 133: If lightning is present the Updraft Source Layer changes relative to the default value of what? Also you don't really need to abbreviate updraft source layer since it doesn't come up again. There are already a lot of acronyms in this paper!

**If lightning is present, the updraft source layer is assigned to be the layer with the greatest moist static energy. As temperature and moisture perturbations are added, this layer does not change. In the default KF scheme, different layers are tested between the surface and the lowest 300 hPa of the atmosphere. If any of them generates an updraft that meets the original KF criteria for deep convection, then that layer is used as the USL. A more thorough description of USL is in Section 2a of Kain (2004). We elected to retain the local acronym "USL" because we use it several times during our description.**

3. Lines 138-151: This paragraph explains why Heath et al changed the criteria for deep convection. The previous paragraph explains that the scheme adds moisture and heat to meet the criteria, but it's a bit confusing since at that point we don't get know what the criteria are and what they have to do with lightning. I suggest switching the paragraph that starts at line 138 with the one that starts at 126 (with other edits to make it flow).

**Thank you for pointing this out. In the updated manuscript, we added the following lines to indicate the specific differences in the LTA method versus the original KF:**

**Lines 143-148: In the unmodified KF scheme, a cloud must exceed a minimum depth (as a function of cloud base temperature) to satisfy the deep convection criteria. Specifically, a cloud base temperature greater than 20°C must have a cloud greater than 4 km deep.  For a cloud base temperature less than 0°C, the**

**cloud depth only needs to be 2 km. For cloud bases between 0 – 20ºC, the minimum cloud depth is defined as 2000 + 100T$_{LCL}$, where T$_{LCL}$ is the temperature at the lifted condensation level (LCL) (Kain 2004).**

**We elected to keep the original order of the paragraphs to align with the calculations are made in the KF scheme itself.**

**References**

Chai, T., and Draxler R. R.: Root mean square error (RMSE) or mean absolute error (MAE)? – Arguments against avoiding RMSE in the literature, *Geosci. Model Dev.*, **7**, 1247–1250, https://doi.org/10.5194/gmd-7-1247-2014, 2014.

Gilliam, R. C., Herwehe, J. A., Bullock, Jr., O. R., Pleim, J. E., Ran, L., Campbell, P. C., and Foroutan, H.: Establishing the suitability of the Model for Prediction Across Scales for global retrospective air quality modeling, *J. Geophys. Res.: Atmos.*, **126**, e2020JD033588. https://doi.org/10.1029/2020JD033588, 2021.

Heath, N. K., Pleim, J. E., Gilliam, R. C., and Kang, D.: A simple lightning assimilation technique for improving retrospective WRF simulations, *J. Adv. Model. Earth Syst.*, **8**, 1806–1824, https://doi.org/10.1002/2016MS000735, 2016.

Hodson, T. O.: Root-mean-square error (RMSE) or mean absolute error (MAE): when to use them or not, *Geosci. Model Dev.*, **15**, 5481–5487, https://doi.org/10.5194/gmd-15-5481-2022, 2022.

Kain, J. S.: The Kain-Fritsch convective parameterization: an update, *J. Appl. Meteorol.*, **43**, 170–181, https://doi.org/10.1175/1520-0450(2004)043<0170:TKCPAU>2.0.CO;2 2004.

Kang, D., Pickering, K., Allen, D., Foley, K., Wong, D., Mathur, R., and Roselle, S.: Simulating lightning NO production in CMAQv5.2: evolution of scientific updates, *Geosci. Model Dev.*, **12**, 3071–3083, https://doi.org/10.5194/gmd-12-3071-2019, 2019.

Willmott, C. J. and Matsuura, K.: Advantages of the mean absolute error (MAE) over the root mean square error (RMSE) in assessing average model performance, *Clim. Res.*, **30**, 79–82, https://doi.org/10.3354/cr030079, 2005.

**Responses to Reviewer 2:**

This manuscript convincingly demonstrates the benefits of lightning data assimilation in the WRF model when run regionally over CONUS and when run over the Northern Hemisphere at courser resolution.  The benefits over CONUS were the greatest when high detection efficiency National Lightning Detection Network data are used, but model performance improvement was noted even when the lower detection efficiency World Wide Lightning Location Network (WWLLN) data are used.  The authors considered the effects of two of the Kain-Fritsch convective scheme parameters (trigger function and convective time step) in association with the lightning assimilation.  These parameters have major effects on precipitation in the base case without lightning assimilation, but the effects of variation of these parametes is diminished when the assimilation is used.   The paper presents comprehensive statistics on the performance of the simulations with the two lightning data sets and with the variation of convective parameters.  The results suggest possible future improvement of the lightning assimilation scheme to take into account horizontal grid resolution by using the observed flash densities to determine when to trigger convection.  The paper certainly fits in the scope of GMD, and I recommend publication after some minor revisions outlined below.

**We thank the reviewer for this overall positive and supportive assessment.**

line 32-33:  monthly mean daily precipitation

**Thanks, the correction has now been made.**

line 64:  add some more references:   Allen et al. (2012); Kang et al. (2019a,b)

**Thank you for the suggestion. The references are now added.**

line 90:  Even though there are some….

**Thanks, the change is now made.**

lnes 93-94:  …there is no literature evaluating how these parameter….

**Thanks, the change is now made.**

line 104:  efficiency is much lower than the >95% of NLDN for cloud-to-ground (CG) flashes

**Thank you so much for catching this point. How have now revised the manuscript accordingly.**

line 109:  ...with NLDN lightning flashes over CONUS

**Thanks, the change is made.**

line 212:   and snow.

**If the "are employed" after "and snow" is removed, the sentence is not complete. So, we keep the sentence as it was written.**

line 218:  move URL to after the word "dataset" in the previous line

**Thanks, the URL is now moved to after the word "dataset".**

line 298:   ...present the more dramatic fluctuations...

**Thanks, the change is made.**

line 376:  ...errors were noticable (Figure 6).

**Thanks for the suggestion. We have now added "Figure 6".**

line 436:  ...among the BASE cases were noted in all the....

**Thanks. We have now revised the sentence as suggested.**

line 462:  In the analysis in Figure 3b....

**Thanks. We now changed the word "with" to "in".**

line 503:  ...12-km LTA cases (both K2C10W and K2C10N)

**Thanks, we have added the case names for clarity.**

line 507:  ...in that the precipitation from Trig2 was....

**Thanks, the change has been made.**

line 574:  ...directions are to use criteria values of lightning flash density dependent on grid resolution to trigger deep convection...

**Thank you so much for the suggestion. We have now revised the sentence as suggested.**

line 586: "updates" Please remind the reader here what the updates were

**Thanks for the suggestion. We have now revised the sentence as: "Separately, the original LTA technique as described in Heath et al. (2016) showed influence from the cumulus parameters on the LTA cases (Figure S8), but after implementing the updated cloud top height (one model level above -20° C) and the additional pre-conditioning shallow convection (see in Section 2), the fluctuations among the LTA cases were significantly reduced."**

line 603: ...the convective processes (e.g., convective transport of air pollutants matching the times and locations of lightning NOx production) to have....

**Thank you. The clarification has now been added.**

line 610: I'm not sure what is meant by "scope" here. Please add "strokes per flash" to this list of new data from GLM.

**Thanks for the comment. We have now replaced the vague word "scope" with "strokes per flash" in the revised sentence.**

lines 640-647: should these items be moved to the "Code and Data Availability" section?

**We agree. We have now moved these descriptions to the "Code and Data Availability" section.**

---

## Author Response (AR2)

Dear authors,

Thanks for your great efforts. Both reviewers are very positive about the revision. I have a minor point that has been mentioned by one of the reviewers. From my point of view, the data assimilation scheme or how to assimilate the lightning data should be clearly stated in the abstract and in the introduction. Otherwise, it does cause some confusion.

Best regards,

Topical editor

GMD

Yuefei Zeng

**Dear Editor Zeng,**

**Thank you very much for your above comments, we have now added the following sentences in Abstract and Introduction, respectively:**

**In Abstract: "*The LTA technique uses lightning data to trigger the Kain-Fritsch convective parameterization via realistic temperature and moisture perturbations.*"**

**In Introduction: "*Specifically, the LTA technique uses temperature and moisture perturbations to trigger KF deep convection where lightning is observed, resulting in a parameterized cloud with realistic characteristics based on the local environment and our understanding of lightning-producing convective clouds.*"**

**Thank you again and hope this revision would lead to the acceptance of this manuscript.**

**Daiwen Kang**
**On behalf of all the co-authors.**